# MiniDeep: A Standalone AI-Edge Platform with a Deep Learning-Based MINI-PC and AI-QSR System

**DOI:** 10.3390/s22165975

**Published:** 2022-08-10

**Authors:** Yuh-Shyan Chen, Kuang-Hung Cheng, Chih-Shun Hsu, Hong-Lun Zhang

**Affiliations:** 1Department of Computer Science and Information Engineering, National Taipei University, No. 151, University Rd., San Shia District, New Taipei City 237, Taiwan; 2Department of Information Management, Shih Hsin University, No. 1, Ln. 17, Sec. 1, Muzha Rd., Wenshan District, Taipei City 116, Taiwan

**Keywords:** MiniDeep, edge computing, deep learning, cloud computing, recommendation system

## Abstract

In this paper, we present a new AI (Artificial Intelligence) edge platform, called “MiniDeep”, which provides a standalone deep learning platform based on the cloud-edge architecture. This AI-Edge platform provides developers with a whole deep learning development environment to set up their deep learning life cycle processes, such as model training, model evaluation, model deployment, model inference, ground truth collecting, data pre-processing, and training data management. To the best of our knowledge, such a whole deep learning development environment has not been built before. MiniDeep uses Amazon Web Services (AWS) as the backend platform of a deep learning tuning management model. In the edge device, the OpenVino enables deep learning inference acceleration at the edge. To perform a deep learning life cycle job, MiniDeep proposes a mini deep life cycle (MDLC) system which is composed of several microservices from the cloud to the edge. MiniDeep provides Train Job Creator (TJC) for training dataset management and the models’ training schedule and Model Packager (MP) for model package management. All of them are based on several AWS cloud services. On the edge device, MiniDeep provides Inference Handler (IH) to handle deep learning inference by hosting RESTful API (Application Programming Interface) requests/responses from the end device. Data Provider (DP) is responsible for ground truth collection and dataset synchronization for the cloud. With the deep learning ability, this paper uses the MiniDeep platform to implement a recommendation system for AI-QSR (Quick Service Restaurant) KIOSK (interactive kiosk) application. AI-QSR uses the MiniDeep platform to train an LSTM (Long Short-Term Memory)-based recommendation system. The LSTM-based recommendation system converts KIOSK UI (User Interface) flow to the flow sequence and performs sequential recommendations with food suggestions. At the end of this paper, the efficiency of the proposed MiniDeep is verified through real experiments. The experiment results have demonstrated that the proposed LSTM-based scheme performs better than the rule-based scheme in terms of purchase hit accuracy, categorical cross-entropy, precision, recall, and F1 score.

## 1. Introduction

Deep learning is a fast growing technology that has attracted lots of attention recently. The deep learning makes a lot of applications feasible. Deep learning has been widely adopted in many fields, such as computer vision, image recognition, speech recognition, and natural language processing.

However, the effectiveness of deep learning is based on the machine computing capability. Deep learning has two main aspects which are relative to computing power, i.e., the training aspect and the inference aspect. The training aspect is an important part to determine the overall outcome of deep learning. As a typical neural network model, deep learning requires training data to learn what the high dimension feature of the dataset is. On the other hand, the inference aspect is the step of putting a deep learning model into the production environment for prediction. The inference aspect relies on the weights of the trained model to provide a prediction of the outcome by passing input to the neural network model and then obtaining the prediction output. Both the training aspect and the inference aspect rely on the computing power. Both of them need a powerful graphics card, which is good at a large number of parallel instruction operations and has enough memory space to perform high-performance data handling.

The other issue is the software dependency with deep learning. In order to use the deep learning model, we need to prepare the entire deep learning software environment, such as TensorFlow, Nvidia CUDA library, Keras, python, etc. This is a barrier when we deploy our applications. We also need to make sure that other libraries work fine. The limitation described above makes the deployment more difficult.

In order to overcome the deep learning hardware restriction and software dependency, an idea that can make deep learning environment stand alone from the local machine is proposed. Using the cloud service to perform deep learning is an easier and effective way to do that. Cloud computing is a distributed system to provide highly scalable and resilient environments. It provides the hardware resource by means of on-demand and moving training, and making inferences on the cloud platform is a solution that is not limited by the hardware computing power. In addition, the software dependency effort will be reduced significantly because all of the software dependency such as the library and environment can be moved to the cloud side. The local machine only needs to handle the software dependency about the application itself.

Although moving the training and inference process to a cloud platform to help our hardware computing demand and software dependency reduction is a good choice, the issue relative to the network behavior needs to be discussed. In principle, we need to send the inference request to the cloud service, triggering the inference process with provided predict input data, obtain the output from the inference result, and then send a response to the local machine, which requests the inference result. In this case, deep learning on the cloud will cause a lot of delays due to network packet exchange. Obtaining a real-time inference response is hard. However, most of the popular deep learning applications, such as face detection, object recognition, semantic segmentation, rely on real-time predictions. They need to upload large amounts of image data to the cloud service, then use a well-known model such as convolution neural network (CNN) to perform the inference processing with many CPU intensive operations and the high-dimension nonlinear calculation going through the CNN model, and finally send the data back to the local device. The latency of inference response is unacceptable for real-time predictions.

The other problem is the loading of the cloud server. Assume that we have a cluster system, which is composed of the central cloud and branches of the local device application. This application will send the inference request to the central cloud. Because all of the applications need to obtain the inference resources from the central cloud, deep learning computing on the cloud will produce a large amount of packets, and the cloud service needs to handle those requests. Deep learning computing on the cloud will cause a huge loading on the cloud service. Even though we can choose higher performance cloud services and more sustainable cloud machines, deep learning computing on the cloud will cost more and more money on this issue.

Deep learning on the edge is a solution for the above issue. Deep learning on the edge can reduce the bandwidth and latency when we send our packet to the nearby edge device. Deep learning on the edge can make it easier for the network to work effectively. Therefore, if we move the inference process from the cloud to the edge, real-time inference through the network becomes feasible.

In summary, edge inference/cloud training is a robotic architecture to achieve the two main goals. One is the standalone ability. This feature resolves the hardware restriction and software dependency on the local machine application by moving the deep learning process away from the local machine. The other is traffic-free; this feature reduces a large amount of traffic effort by using the edge device as the nearest computing resource. Therefore, building a standalone and traffic-free platform is necessary. In this paper, we provide developers with a whole deep learning development environment called MiniDeep to set up their deep learning life cycle processes. To the best of our knowledge, the existing AI-Edge platforms [1,2,3] do not provide a whole deep learning development environment for the developers or just provide a framework for the development of the deep learning system [4].

The contributions of the proposed scheme are listed as follows.

We have proposed and built a new AI-Edge platform called “MiniDeep”.The proposed MiniDeep can provide developers with a whole deep learning development environment to set up their deep learning life cycle processes, such as model training, model evaluation, model deployment, model inference, ground truth collecting, data pre-processing, and training data management. To the best of our knowledge, such a whole deep learning development environment has not been built before.The proposed MiniDeep can use a stand-alone mini-size PC as the edge device. Hence, it can be deployed on the nearby local machine easily, and it has the ability to provide a plug and play feature.In the proposed platform, a cloud-edge based deep learning software system can use the edge device to perform inference and use the cloud service to train the neural network model.A recommendation system for AI-QSR (Quick Service Restaurant) KIOSK (interactive kiosk) application has been implemented on the proposed platform. The experiment results have demonstrated the effectiveness of this recommendation system.

The remainder of this paper is organized as follows. Section 2 describes the related works and motivations. Section 3 describes the system architecture, the basic idea, and system components of our platform. Section 4 describes the MiniDeep system architecture. Section 5 introduces the AI-QSR system architecture. Section 6 provides the experimental results. The conclusion is finally given in Section 7.

## 2. Related Work

### 2.1. Literature Review

Jacob et al. [5] use Raspberry Pi 3 plug with an Intel Neural Compute Stick to perform real-time object recognition for vehicular edge computing. The paper shows that an embedded system such as Raspberry Pi 3 with an Intel Neural Compute Stick is capable of performing real-time object recognition by edge computing. Jaewon et al. [6] propose a new design of the Cloud-Edge Collaboration framework for IoT data analytics. The platform uses the crawler to obtain the IoT training data and training model on the cloud, and they select the best performance model to deploy on the edge device. The edge device considers the inference result and control the device by the interaction rule. Hongxing et al. [7] propose a platform “WiCloud” to provide edge networking, proximate computing and data acquisition for innovative services. Nam et al. [1] propose a novel edge computing platform, Distributed-Node-RED, that uses a distributed data flow programming model based on an open-source Node-RED tool. They address challenges for building fog application across the edge network to the cloud. Preethi et al. [8] introduce a sentiment analysis recommendation system based on Recursive Neural Networks (RNN); the sentiment analysis is performed on the different reviews, which are obtained from different social media. Yanik et al. [2] introduce the edge computing platform for smart building, names Qarnot architecture, which locally builds and calibrates smart-building systems for processing of the natural acoustic flow. Shangyu et al. [9] present a resource allocation scheme call Mobile-I, which optimizes both computing and storage resources so that the processing delay can be reduced and the resource can be utilized on the edge device. Velasco-Montero et al. [10] compare different deep learning models on the embedded edge device, and they define a Figure of Merit (FoM) that merges the performance metrics into a representation of computational and energy efficiency models and frameworks on the hardware platform. Jonghwan et al. [11] introduce the feature space encoding, which performs the inference to the middle layer; then, they encode the feature and transfer it from the edge device to the local device to complete the subsequent inference process. They also provide the method of compressed (loss-less and lossy) for feature space encoding before transmitting to the local device. Chunwei et al. [12] test different DNN models on different paltforms. The result shows that modern edge CPUs are suitable for inference with DNN models such as LSTM, MLP, and mobile-optimized CNNs. Rothe et al. [13] propose a state-of-art age recognition by considering the age estimation problem as a fused of regression and classification combination. The used network is VGG-16, which is classified by a pre-training model on ImageNet and then trained on the IMDB-WIKI dataset. The contribution of re-processing methods is that the face feature point extraction is not used in this article. Sergey et al. [14] present the Wide Residual Networks, which is a model that is used to more effectively improve the performance of the model. The authors make the model wider by increasing the number of output channels, so that network depth N can maintain a small value, and the network can achieve good results. Jiaxi et al. [15] propose a Convolutional Sequence Embedding Recommendation Model (Caser). They find that it is important that the sequential patterns have a larger effect on the next item. Cheng et al. [16] propose a matrix factorization method, namely FPMCLR, to embed the personalized Markov chains and the localized regions. The experimental results show that is effective and efficient, and they compared it to several state-of-the-art methods to prove the performance. Ruining et al. [17] propose a new method, called TransRec, for large-scale sequential prediction. They indicate a “transition space” that is embedded from items, and the user is represented as translation vectors operating on item sequences. This approach has good performance in real-world datasets. Balázs Hidasi et al. [18] apply RNN to the Session-based Recommendation for the first time, and they design the RNN training, evaluation method and ranking loss for this task. Balázs Hidasi et al. [19] explore how to add the item property information such as text and images to the RNN framework, and they explore several model frameworks that incorporate item properties. Massimo et al. [20] propose a hierarchical RNN model. Compared with the previous work, they can describe the personal interest changes of the users in the session and make the user’s personalized session recommendation. Jannach et al. [21] show that the RNN model in the session can be combined with the KNN method to improve the recommendation effect. Tossawat et al. [22] propose a recommendation system which uses a deep learning model to extract ingredients from the user’s favorite dishes, and it also collects the history of dish-selecting information with a user profile in a database. Pui et al. [23] develop a framework for medical KIOSK call MedKIOSK, which has the ability to perform intelligent conversation with the user by deep learning with natural language processing. Wang et al. [24] propose a heterogeneous brain storming (HBS) online knowledge distillation (KD) scheme for object recognition in real-world Internet of Things (IoT) scenarios. This scheme converts the traditional temperature hyperparameter to a trainable parameter to find optimal temperature for different networks, tasks, and methods. Kasi et al. [25] address the edge server placement problem within an existing network infrastructure. The problem of edge server placement is formulated as a multiobjective constraint optimization problem to balance between the workloads of edge servers and reduce access delay between the base stations and edge servers. The genetic algorithm and local search algorithms are adopted to find the best solution. Riggio et al. [4] describe the challenges and conceptual architecture of the AI@EDGE project. The AI@EDGE project aims to provide a flexible AI/ML (Artificial Intelligence/Machine Learning) platform at the network edges where the next generation AI-enabled application and services can be deployed. Choi et al. [3] present real-time object detection models for AI edge platforms using a modified head structure of the refinement detector (RefineDet). Through performance tests and model analysis, a balanced performance in terms of accuracy and speed for real-time detection on edge platforms can be achieved.

### 2.2. Motivations

Through the above discussion, we can summarize that edge computing and cloud-edge collaboration can be adopted to applications such as inference and object recognition. Many AI-based recommendation systems have been proposed and proven to be effective. However, the AI-based recommendation systems have not been built on an AI-Edge platform. To build an AI-based recommendation system on the AI-Edge platform, we have proposed a novel AI-Edge platform called “MiniDeep”. MiniDeep provides developers with a whole deep learning development environment to set up their deep learning life cycle processes. With MiniDeep, developers can build an AI-based edge-cloud system more easily. Although many AI-Edge platforms have been proposed [1,2,3], these AI-Edge platforms do not provide developers with a whole deep learning development environment. The AI@EDGE project [4] aims to provide a flexible AI/ML platform at the network edges where the next generation AI-enabled application and services can be deployed. However, the AI@EDGE project has just provided a framework for the development of the deep learning system. The whole deep learning development environment has not been built yet.

## 3. Preliminaries

This section describes the knowledge of the edge-cloud architecture and introduces the cloud service and edge environment that are used in this paper.

### 3.1. Edge-Cloud Architecture for Deep Learning

Typically, a deep learning life cycle contains many detailed procedures. There are several procedures in a life cycle such as data collecting, data pre-possessing, model training, model evaluation, model deployment, inference monitoring, and starting a new life cycle by going back to data collecting. Implementing the deep learning life cycle on the production machine may incur challenges because deep learning needs to compute tons of neural forward operations to obtain the model inference output. Training a model on a production machine is also a very difficult solution because of the poor hardware computing power on the production machine. In summary, separating the computing resource of the production machine from actual deep learning computing is a good choice to prevent deep learning computation from occupying the whole computing power of the production machine. In this paper, we propose a standalone deep learning platform in mini pc. Mini pc is based on the edge-cloud architecture and has the ability to train and inference a deep learning model for the production machine.

The system architecture of the edge-cloud model for deep learning is shown in Figure 1. The system architecture is divided into two major aspects, the cloud side and the edge side. The cloud side is based on the machine learning services such as AWS sagemaker, Google Cloud ML Engine, Azure Machine Learning, IBM Watson Machine Learning, etc.

The machine learning service gives the developer a deep learning developing environment to train a deep learning model with various computing hardware instances. The machine learning service generally provides the script coding, model training, hyperparameter tuning, and model storing when developing on the cloud.

The remaining part of the cloud side is the continuous training management and AI deployment service. The continuous training management is a process to manage model training scripts and schedule the period to obtain training data and perform online training automatically. Deploying a deep learning service on the edge device needs to rely on the AI deployment service. The AI deployment service provides the ability to deploy a trained model automatically to the edge device when the deep learning model has been trained successfully.

The edge side is based on the embedded system device or the PC for deep learning inference. The edge side uses the computing hardware to compute the deep learning model inference. There are many options for choosing computing hardware, such as an embedded system board, AI Chips, AI USB, etc. An Nvidia Jeston TX2 embedded board has a dual-core NVIDIA Denver2, a quad-core ARM Cortex-A57 CPU, and a 256-core Pascal GPU. Nvidia Jeston TX2 is suitable for developing deep learning inference work on the edge side. The other choice is to use mini pc with an external computing resource. Using an Intel neural compute stick is a good choice for the edge device whose computing power is not enough to cope with the deep learning inference.

The AI acceleration platform on the edge side is an environment that helps the edge device to enable hardware computing acceleration. The edge device acceleration platform has many optimizing operations to accelerate the deep learning inference, such as parallel computing, heterogeneous calculation, neural network optimization, etc. OpenVino enables CNN-based deep learning inference at the edge and supports heterogeneous execution across computer vision accelerators [26].

The remaining part of the edge side is the data synchronizer and deployed environment. The data synchronizer is a local data storage management system which has the ability to handle the data uploading process from the edge to the cloud. The data synchronizer also provides the definition of the pre-processing process for new data collected from the inference. The deployed environment is a model deployment and model inference execution environment on the edge device. This environment is often based on the AI acceleration platform to perform deep learning inference. This environment also needs to provide a restful API interface so that the local application device can send a request and obtain a response from the deep learning inference to the edge device.

### 3.2. Deep Learning Cloud Service Overview

In this paper, MiniDeep uses Amazon Web Service (AWS) as a deep learning model management platform. MiniDeep uses AWS to handle some deep learning jobs, such as data storing, model training, model evaluation, and model deployment. Sagemaker is one of the AWS that provides a platform to host a deep learning runtime environment. The runtime environment can train the model and evaluate the model with the uploading model script from the developer. Sagemaker provides the training hardware which you need so that the developer can only focus on the deep learning operation without worrying about hardware maintenance. AWS IoT GreenGrass is one of the AWS that can bring AWS services to the edge device. GreenGrass can choose the serverless instance with user scripts and the package serverless instance as a GreenGrass core which can be deployed to the edge device. With this feature, GreenGrass can provide a solution to package the deep learning model to the edge and keep tracking the edge device status by the GreenGrass core feature.

The above two services are the main services which MiniDeep deep learning operations are based on. Other than this, the basic serverless service as a platform module is needed to be used to implement the core function. MiniDeep uses AWS Lambda service to host a serverless runtime to compute the application logic by demand. Using the AWS lambda service can only focus on what system logic you care about, and you do not need to care about what infrastructure and operation environment you used. For data storing, MiniDeep uses AWS DynamoDB to support the platform for storing the training data uploaded from the edge side. With AWS Lambda service and DynamoDB service, MiniDeep can build a suitable serverless architecture with our system.

### 3.3. Deep Learning Edge Environment Overview

This section introduces the runtime environment which is built in the MiniDeep edge device. The edge device uses Intel OpenVino as the hardware acceleration platform so that MiniDeep can make inferences more efficient by hardware acceleration. We know that OpenVino has its limitations. However, we aim to use a non-Nvidia (mainstream) architecture. We want to confirm that using OpenVino plus Intel Movidius is more suitable for an environment with small space, no fan, and less maintenance. The Intel OpenVino is used in the CNN-based deep learning inference at the edge, and it supports heterogeneous execution on the hardware resource of the edge device. With the acceleration ability, OpenVino executes deep learning acceleration on CPU, GPU, Intel Movidius Neural Compute Stick, and FPGA. OpenVino is composed of the Model optimizer and Inference engine.

A model optimizer is a command-line tool that performs static model analysis and adjusts deep learning models for the optimal execution on edge devices. After that, the model optimizer produces an Intermediate Representation (IR) of the network, which can be loaded with the Inference Engine.

The Inference Engine is a C++ library with a set of C++ classes to obtain the inference result. The C++ library provides an API to read IR which is obtained from the model optimizer. The input and output formats are set, and the inference result on edge devices is executed. The Inference Engine allows high-performance inference by paralleling execution and heterogeneous execution on many hardware types including Intel CPU, GPU, FPGA, and an Intel Movidius Neural Compute Stick.

In order to accelerate with the help of external hardware, the Intel Movidius Neural Compute Stick is plugged into the MiniDeep edge device. With Neural Compute Stick (NCS), Intel NCS can help to accelerate the inference for poor hardware computing devices. NCS may provide an environment with fast response inference for real-time applications.

### 3.4. Problem Definition

In this paper, there is an AI-QSR (Artificial Intelligence Quick Service Restaurant) KIOSK (interactive kiosk) application based on the MiniDeep platform. The AI-QSR has a recommendation system to recommend to the end-users which food they may like to purchase. The goal of the recommendation system aims to provide a high purchase intention QSR recommendation service.

The food item is the basic element in the AI-QSR system. The composition of the multiple food items can be a combo suit. For example, one hamburger food item and drink food item can be a basic hamburger combo suit. The AI-QSR has a menu category where each element in the menu category is a food item or a food combo suit. In this case, all the food items in AI-QSR can be represented as a set $F=(f_{1},...,f_{i},...,f_{n})$, where *n* is the total number of food items in the AI-QSR application. For each food item $f_{i}$ in *F*, $f_{i}$ contains the information of the food feature. Each food item $f_{i}$ is represented as $f_{i}=\left\{f_{i}^{p},f_{i}^{ca},f_{i}^{m},f_{i}^{co}\right\}$. $f_{i}^{p}$ represents the food price, where $f_{i}^{p}=\left\{N^{+}\right\}$. $f_{i}^{ca}$ is the number of food item categories, where $f_{i}^{ca}=\left\{0,1,...,c-1\right\}$ and *c* is the number of categories. $f_{i}^{m}$ represents the number of menus, where $f_{i}^{m}=\left\{0,1,...,m-1\right\}$ and *m* is the number of menus. $f_{i}^{co}$ represents a food item content vector with six elements, where $f_{i}^{co}=\left(c_{1}^{v},c_{2}^{v},...,c_{6}^{v}\right)$ and $c_{k}^{v}$ is a content vector element. For each content vector element in $f_{i}^{co}$, the content vector element represents a hamburger, chicken, drink, snack, sandwich, and suit, respectively.

The user’s information data contain the important descriptions to represent a user’s feature, which can be obtained from a camera, user interface, or usage flow. For each user $u_{i}$ in user set *U*, there is a $u_{i}=\left\{d_{i}^{a},d_{i}^{g},d_{i}^{c},d_{i}^{p}\right\}$. $d_{i}^{a}$ is the user age scalar number, where $d_{i}^{a}=\left\{N^{+}\right\}$. $d_{i}^{g}$ is the user gender binary data, where $d_{i}^{g}=\left\{0,1\right\}$. $d_{i}^{c}$ is the user click event which is a variable-length vector, where $d_{i}^{c}=\left(e_{1}^{i},...,e_{j}^{i},...,e_{t}^{i}\right)$ and *t* is the time step of the most current click event, and each click event $e_{j}^{i}$ is the positive number given by $e_{j}^{i}=\left\{N^{+}\right\}$. The click event $e_{j}^{i}$ is a flow sequence element which represents a click step of user usage flow. $d_{i}^{c}$ is a whole flow sequence which represents a whole step of purchase flow generated by the user. $d_{i}^{p}$ is a list of food items that contains the food item purchased by the user previously. $d_{i}^{p}$ is represented as $d_{i}^{p}=\left(p_{1}^{i},...,p_{j}^{i},...,p_{k}^{i}\right)$, where *k* is the total number of food items purchased by the user. Each purchased item is a food item given by $p_{j}^{i}\in F=\left(f_{1},f_{2},...,f_{n}\right)$

For the symbol representation of food item $f_{i}$ and user information $u_{i}$, there is a recommendation accuracy problem in QSR-FS (Quick Service Restaurant Flow Sequence). In the QSR-FS problem, AI-QSR KIOSK application needs to maximize the recommendation accuracy. The higher recommendation accuracy can be considered as the higher user purchase intention. Once this problem has been resolved, the KIOSK application with a recommendation system for food suggestion can be widely used in a quick service restaurant.

### 3.5. Problem Formulation

The QSR-FS problem described in the previous paragraph can be formulated as follows.
(1)$$\begin{array}{c} A\approx \underset{\theta^{*}}{\arg\max}\left\{QSR\_FS(\theta^{*})\right\} \end{array}$$

The purpose of Equation (Equation 1) is to find the highest accuracy *A* in the QSR-FS problem. $\theta^{*}$ is the output of the loss function which can be found by the given θ in Equation (Equation 2). The loss function tries to find a θ which minimizes the output of the loss function to fit with the dataset. The loss function is used to estimate the degree of inconsistency between the predicted value f(x,θ) of the model and the true value *y* that is a non-negative real-valued function, which is usually represented as L(Y,f(X)). The smaller the loss function is, the better the robustness of the model becomes. There is a representation *X* for the input data of the QSR-FS formula, as shown in Equation (Equation 3), and *Y* for the labeled data of the QSR-FS formula, as shown in Equation (Equation 4). In this case, *r* is the number of training data in *X*, and *s* is the number of purchased food item in one training data.
(2)$$\begin{array}{c} \theta^{*}=\underset{\theta }{\arg\min}\left\{\frac{1}{N}\cdot \sum_{i=1}^{n}\cdot L\left(y_{i},f\left(x_{i},\theta \right)\right)+\lambda \Phi \left(\theta \right)\right\}\;,\;where\left\{\begin{array}{c} X=(x_{0},\cdots ,x_{i},\cdots ,x_{n})\\ Y=(y_{0},\cdots ,y_{i},\cdots ,y_{n}) \end{array}\right. \end{array}$$
(3)$$\begin{array}{c} x_{i}\in \left\{u_{1},\cdots ,u_{i},\cdots ,u_{r}\right\},\;subject\;to\left\{\begin{array}{c} u_{i}=\left\{d_{i}^{a},d_{i}^{g},d_{i}^{c},\left(f_{1},\cdots ,f_{j},\cdots ,f_{s}\right)\right\},\\ \;f_{j}=\left\{f_{j}^{p},f_{j}^{ca},f_{j}^{m},f_{j}^{co}\right\} \end{array}\right.\\ \; \end{array}$$
(4)$$\begin{array}{c} y_{i}\in \left\{f_{1},...,f_{i},...f_{r}\right\} \end{array}$$

In Equation (Equation 2), $L(y_{i},\hat{y_{i}})$ is the category cross-entropy function to classify multiple cases of output, as shown in Equation (Equation 5). There are also the regularization Φ and the regularization constant λ that Φ can be placed as L1 regularization and L2 regularization, as shown in Equations (Equation 6) and (Equation 7), respectively. The regularization is the part to prevent over-fitting and improve the generalization ability in the loss function.
(5)$$\begin{array}{c} L\left(y_{i},\hat{y_{i}}\right)=-\sum_{j=1}^{m}\hat{y_{ij}}\log y_{ij}. \end{array}$$
(6)$$\begin{array}{c} \Phi_{L1}\left(\theta \right)=\sum_{i=1}^{n}∥\theta_{i}∥. \end{array}$$
(7)$$\begin{array}{c} \Phi_{L2}\left(\theta \right)=\sum_{i=1}^{n}\theta_{i}^{2}. \end{array}$$

Furthermore, Equation (Equation 8) is the QSR-FS function which uses exponential function with $-\theta^{*}$. This equation obtains the cross-entropy result of the loss function to obtain the approximate accuracy *A*. The cross-entropy is the probability of being correctly classified for a single sample. When the learning algorithms of the training machines are using batch-based and cross-entropy loss, the accuracy can be roughly calculated according to loss, and the error decreases as the batch size increases. When the batch size is set as 100—that is, between e-loss and accuracy–the error is usually less than 0.01.
(8)$$\begin{array}{c} QSR\_FS\left(\theta^{*}\right)\approx e^{-\theta^{*}} \end{array}$$

Finally, by combining all the above formulas, the QSR-FS problem can be formulated as shown in Equation (Equation 9).
(9)$$\begin{array}{c} A\approx \underset{\theta^{*}}{\arg\max}\left\{e^{-\underset{\theta }{\arg\min}\left\{\frac{1}{N}\cdot \sum_{i=1}^{n}\cdot L\left(y_{i},f\left(x_{i},\theta \right)\right)+\lambda \Phi \left(\theta \right)\right\}}\right\}\\ \;,\;subject\;to\left\{\begin{array}{c} X=(x_{0},\cdots ,x_{i},\cdots ,x_{n})\\ Y=(y_{0},\cdots ,y_{i},\cdots ,y_{n}) \end{array}\right.\\ \; \end{array}$$

## 4. MiniDeep Edge-Cloud Platform

In this section, the MiniDeep system architecture is described in Section 4.1 to realize the detailed module inside the MiniDeep platform. The training procedure and details of the inference procedure are described in Section 4.2.

### 4.1. Minideep System Architecture

Figure 2 shows the internal architecture of the MiniDeep platform, which is concatenated with the end-device system. The KIOSK application is used as an end-device system. The MiniDeep’s deep learning provider module is installed, which contains the GreenGrass IoT device SDK to call the GreenGrass core function and obtain access with the MiniDeep edge device. The edge device is set at the nearby end device connecting with the Ethernet cable. Communication can be accomplished by the HTTP protocol. The system contains four microservice modules, i.e., Train Job Creator (TJC), Model Deployer (MD), Inference Engine Handler (IEH), and Data Provider (DP). The first and second microservice modules are on the cloud. The third and last microservice modules are on an edge device.

In the MiniDeep system, deep learning model management relies on cloud microservice modules, i.e., Train Job Creator and Model Deployer. The MiniDeep user uploads their model handling script on Train Job Creator. The model handling script contains the procedure of how the model will be trained, the model will be evaluated, and the dataset will be used. The dataset can be provided by Data Provider, which collects the ground truth data on the end device and synchronizes to Train Job Creator or is provided directly by DeepLearning Provider SDK, which sends the ground truth data to Train Job Creator directly. After the MiniDeep user uploads their model handling script, the MiniDeep can start model training and model evaluation by using the MiniDeep management tool. The model deployer can take a trained model from the Train Job Creator and package model as a serverless runtime function base on the AWS lambda service. The MiniDeep user can choose which model they want to package and how many of the different models they want to bring together.

After the MiniDeep user finishes their model tuning work on the cloud, the model can be pulled from the Model Deployer to the Inference Engine Handler in the edge device by the MiniDeep management tool. Then, the Inference Engine Handler obtains the model that is packaged as GreenGrass core and then deploy GreenGrass core in the runtime environment of the Inference Engine Handler. The end device can send the RESTful API request to the edge device and then obtain the inference result from the edge device.

When the end device obtains the inference result, the edge device can also obtain the ground truth from the correct feedback of the end-user in the application. The end-device application can send back the correct label of the user as the new training data to the MiniDeep system. The MiniDeep user uses the MiniDeep management tool to upload the new training data to the Data Provider on the edge device or send the new training data directly to the Train Job Creator dataset database. The Data Provider itself also manages the new training data and synchronizes the new training data by mechanisms between the Data Provider and Train Job Creator.

### 4.2. Minideep Platform Usage Design

This section describes how the individual microservice module works in the training and inference phases. This section also describes how the microservice modules co-work together to upload training data and deploy the inference model by our protocol.

The detailed functions of the Data Provider (DP) and Train Job Creator (TJC) are described as follows.

#### 4.2.1. Data Provider

Figure 3 shows the detailed functions of Data Provider (DP). The functionality of Data Provider is to store the new training data in the edge device temporarily and then send the stored new training data to the cloud service if the cloud service requests Data Provider. There are two core services and one management component. The two core services are the data_collection_service and training_dataset_service.

The training_dataset_management component is a MongoDB-based training database. The component, which is called the training_dataset_management component, stores the data uploaded from the local device, manages data as a JSON format data in the database, and then returns the unsynchronized training data to the Train Job Creator. The training_dataset_management component represents the dataset as $D_{i}^{\hat{d}}$, one training datum as $d_{i}^{\hat{d}}$, one feature datum as $x_{i}^{\hat{d}}$, and one label datum as $y_{i}^{\hat{d}}$. The data stored in the training_dataset_management can be represented as follows:(10)$$D^{\hat{d}}=\overset{n}{\sum_{i=1}}D_{i}^{\hat{d}}=\overset{m}{\sum_{j=1}}\overset{n}{\sum_{i=1}}d_{\left(i,ji\right)}^{\hat{d}}=\overset{m}{\sum_{j=1}}\overset{n}{\sum_{i=1}}\left(x_{\left(i,ji\right)}^{\hat{d}},y_{\left(i,ji\right)}^{\hat{d}}\right)$$

The data stored in training dataset management can also be divided into two parts, synchronized training dataset $D_{i}^{\hat{d}\_S}$ and unsynchronized training dataset $D_{i}^{\hat{d}\_U}$, which are represented as follows:(11)$$D^{\hat{d}}=D_{i}^{\hat{d}\_S}+D_{i}^{\hat{d}\_U}$$

The data_collection_service is a RESTful endpoint which listens to the HTTP connection and accepts post requests from the local device which sends a new training datum $\left(x_{\left(i,ji\right)}^{\hat{d}},y_{\left(i,ji\right)}^{\hat{d}}\right)$. The service called data_collection_service uses the data upload endpoint to obtain the new training data from the outside and uses a database controller to control training data storage on training dataset management. In addition, the Data Provider has a dataset status checker function that handles the dataset status report. The data_collection_service asks the data_collection_service and sends back to the local device after the data_collection_service obtains the dataset status.

The training_dataset_service is also a RESTful endpoint which listens to the HTTP connection and accepts the HTTP GET request to obtain the unsynchronized training data $D_{i}^{\hat{d}\_U}$ and upload to the Train Job Creator. This service contains a data retrieve function, a batch queue, and a training dataset updater. The Train Job Creator will take the training dataset updater as an endpoint entry and request the new unsynchronized training data $D_{i}^{\hat{d}\_U}$. The training dataset updater checks the batch queue to find if there are enough new data or not. If the batch queue has enough new data $D_{i}^{\hat{d}\_U}$, the training dataset updater will pop the new data and download new data to Train Job Creator to store and train it. The data retrieve function can ask the database controller to obtain new data and push new data in the batch queue every time when the training_dataset_service accepts a connection.

#### 4.2.2. Train Job Creator (TJC)

After obtaining the training data from the edge, the TJC can manage the training data, training script, training job, and model artifacts on the cloud. Figure 4 shows the detailed functions of TJC. There are five lambda-based core services, such as data_synchronize_trigger_service, event_schedule_service, model_instance_launch_service, training_status_handle_service, and the model_artifact_service. Each service is also a serverless microservice in which if one core service does not work, other microservices will not be affected and still work independently. The core services of TJC operating resource elements are managed by management instances, such as dataset_management. The management is called model_script_management and model_artifact_ management. The management instances are based on the AWS DynamoDB or Simple storage service. The end device or edge device can communicate with the Train Job Creator by MiniDeep SDK or the MiniDeep cloud interface, respectively.

The dataset_management is used to manage the training dataset. The training dataset *T* is given by:(12)$$T=\{\left(D_{1}^{\hat{t}},r_{1}^{\hat{t}},i_{1}^{\hat{t}}\right),\left(D_{2}^{\hat{t}},r_{2}^{\hat{t}},i_{2}^{\hat{t}}\right),...,\left(D_{n}^{\hat{t}},r_{n}^{\hat{t}},i_{n}^{\hat{t}}\right)\},$$
where the dataset database *D* is represented as $D_{i}^{\hat{t}}=\left(d_{\left(i,1\right)}^{\hat{t}},d_{\left(i,2\right)}^{\hat{t}},...,d_{\left(i,j\right)}^{\hat{t}}\right)$, the route table of the dataset $R^{\hat{t}}$ is represented as $R^{\hat{t}}=\left(r_{1}^{\hat{t}},r_{2}^{\hat{t}},...,r_{i}^{\hat{t}}\right)$, and the data identification record $I^{\hat{t}}$ is represented as $I^{\hat{t}}=\left(i_{1}^{\hat{t}},i_{2}^{\hat{t}},...,i_{i}^{\hat{t}}\right)$. Each dataset database element $D_{i}^{\hat{t}}$ represents a training data composed of feature data $X^{i}$ and label data $Y^{i}$. For each dataset, the database element $D_{i}^{\hat{t}}$ is given by,
(13)$$D_{i}^{\hat{t}}=\left\{\begin{array}{cc} X^{i}, & where\;X^{i}=\left(x_{1}^{i},x_{2}^{i},...,x_{n}^{i}\right)\\ Y^{i}, & where\;Y^{i}=\left(y_{1}^{i},y_{2}^{i},...,y_{n}^{i}\right) \end{array}\right.\;$$

The route table of the dataset is used to record what model script is used in training the database $D_{i}^{\hat{t}}$. For Equations (Equation 12) and (Equation 13), there is a training dataset complete representation:(14)$$T=\{\left(\left(X^{1},Y^{1}\right),r_{1}^{\hat{t}},i_{1}^{\hat{t}}\right),\left(\left(X^{2},Y^{2}\right),r_{2}^{\hat{t}},i_{2}^{\hat{t}}\right),...,\left(\left(X^{n},Y^{n}\right),r_{n}^{\hat{t}},i_{n}^{\hat{t}}\right)\}$$

When MiniDeep manages the deep learning model training, MiniDeep needs the model script to know how to train deep learning model for the dataset by the procedure written in the model script. The model script is managed by the model_script_management component. The model_script_management component uses simple storage service as a backend service to handle model script storing, and the model script $S^{\hat{t}}$ in model_script_management component is represented as:(15)$$S^{\hat{t}}=\left\{s_{1}^{\hat{t}},s_{2}^{\hat{t}},\cdots ,s_{n}^{\hat{t}}\right\}$$

Each model script $S^{\hat{t}}$ in the model_script_management component has the script itself and the script meta-data. The script meta-data defines the type of the deep learning algorithm and the application of this script. The script meta-data also record the deep learning hyperparameters to define how to train this script: by which epochs, batch size, input dimension, output dimension, and so on. The model script $S^{\hat{t}}$ is also recorded in the dataset_management component. The route table of the dataset $R^{\hat{t}}$ of the dataset_management component records the model script and the used dataset to perform model training. The one dataset route table $R^{\hat{t}}$ may record many model scripts $S^{\hat{t}}$ such that multiple training scripts use the same dataset to do similar things. In summary, the one dataset route table $r_{i}^{\hat{t}}$ in $R^{\hat{t}}$ is given by:(16)$$r_{i}^{\hat{t}}\in \left\{s_{1}^{\hat{t}},\cdots ,s_{i}^{\hat{t}},\cdots ,s_{n}^{\hat{t}}\right\},\;where\;s_{i}^{\hat{t}}\in S^{\hat{t}}$$

With this dataset route table $R^{\hat{t}}$, MiniDeep can find the model script $S^{\hat{t}}$, so the training trigger can invoke the training schedule when the training script request is arranged or called. The data_synchronize_trigger_service is a service that manages the triggers to control when to collect new training data on the edge device and when to start the training job on the cloud by scheduling. This data_synchronize_trigger_service contains a trigger route controller and a data synchronize handler. The trigger route controller has the function to handle how to obtain the dataset to feed new data in dataset_management and obtain the script to find the dataset and launch dataset. The data synchronize handler has the function to synchronize the new training data from the edge side when the trigger is announced. This data_synchronize_trigger_service is placed in the lambda service to serve as a microservice.

The data_synchronize_trigger_service has a lambda runtime which hosts a trigger route controller. The controller uses a device_trigger table to find which trigger is needed to forward to which device. The service handles the new trigger event to perform data fetching from the end device, and the trigger finds the end device De_id and assigns a trigger to the data_synchronize_handler to obtain the new training data, and then, it stores the new training data to dataset_management.

The event schedule service has the ability to set an event rule on the CloudWatch service. This service provides an interface to set an event, and the event is used to announce the service which is called the data_synchronize_trigger_service to obtain the new training data and to launch the training script $S^{\hat{t}}$. This service contains an event controller as the host. The event set component is the event rule to handle which CloudWatch trigger rule should be announced. The event controller has the HTTP endpoint to obtain the edge device message Event_setting (event_type,event_opt). CloudWatch is an AWS service which provides the service to set the event emitter and track the metric to know when to emit. The event setting component has the sync trigger to be a trigger endpoint for the MiniDeep system. The event setting message decides the event type and event opt; then, the event controller can find the correspond sync trigger. Then, the sync trigger can set the AWS CloudWatch trigger according to the event setting. The event schedule service is placed in the lambda service to serve as a microservice.

If MiniDeep wants to manage training script $S^{\hat{t}}$, the model instance’s launch service is a core function to handle this demand. This service provides the API interface for the MiniDeep user to control the train model script $S^{\hat{t}}$. The model instance’s launch service contains the model script controller, job queue, and the instance launcher functions. The model script controller handles the whole life cycle and operation on a training job. The model script controller can control all of the training jobs in the training_job_management_service. The job queue is a queue implementation which arranges the training launch requests in a queue. The instance launcher is a function to start the training script training life cycle and handle the training dataset feeding, hyperparameters forwarding, and job metadata checking. The model instance launch service is placed in the lambda service to serve as a microservice.

The training job management is a component that provides the training job environment with the full management of the service runtime. This management service is based on the sagemaker runtime, and it provides the setting option to define the hardware acceleration solution being applied on the training job. The training job is represented as a training script process which provides the sagemaker runtime to know what model architecture needs to be trained and how to train a deep learning model by the method written in the script. The training job $t_{i}^{\hat{t}}$ is followed with the training instance and metrics; one is the training hardware, the other is the model analysis metrics that the MiniDeep can obtain by the training_status_handle_service.

The training job $t_{i}^{\hat{t}}$ is represented as below:(17)$$t_{i}^{\hat{t}}=\left(s_{i}^{\hat{t}},t_{i}^{\hat{t}\_h},t_{i}^{\hat{t}\_m},a_{i}^{\hat{t}}\right),\;where\;s_{i}^{\hat{t}}\in S^{\hat{t}}$$

Equation (Equation 17) shows the training instance content. The training instance contains a training job, where $s_{i}^{\hat{t}}$ is the model script, $t_{i}^{\hat{t}\_h}$ is the model hardware training instance, $t_{i}^{\hat{t}\_m}$ is the model metrics, and $a_{i}^{\hat{t}}$ is the model artifact. The model artifact is the model training product after the training is completed. The model is stored in the Model artifact management component.

The training_status_handles_service provides an API that the MiniDeep user can use to obtain the training job status, training log, training metrics, or set the resulting trigger so that the MiniDeep can obtain the result event when the MiniDeep user sets the result trigger callback. The training_status_handle_service contains an instance result trigger and model status publisher. The instance trigger is an event handler which listens to the model train job status and records d_set in an instance. If the training job $t_{i}^{\hat{t}}$ is completed, failed, or suspended, the instance result trigger can announce an event to trigger the model status publisher. The model status publisher provides the function to publish the event to the edge side. If the instance result trigger announces the trigger, the model status publisher catches the result and sends the event data back to the edge device according to De_id and trigger reference. The training_status_handle_service is placed in the lambda service to serve as a microservice.

The model artifact management is a component that manages the model artifact $a_{i}^{\hat{t}}$ in the MiniDeep platform. The model artifact $a_{i}^{\hat{t}}$ is the deep learning model product that is produced by the sagemaker service training instance. The model artifact can be treated as a deep learning model product and applied to the deep learning inference environment. The model artifact management contains many artifact elements. Each element contains one model $a_{i}^{\hat{t}\_m}$ and one model meta $a_{i}^{\hat{t}\_mt}$. The model $a_{i}^{\hat{t}\_m}$ is a product available model instance, which can be loaded in the deep learning framework and produce the inference result by predicting data. The model meta $a_{i}^{\hat{t}\_mt}$ is a file that records the whole model training log, model status, model training metrics. The model artifact $a_{i}^{\hat{t}}$ can be retrieved by model_artifact_service. The model artifact element $a_{i}^{\hat{t}}$ in model_artifact_management_service can be represented as:(18)$$a_{i}^{\hat{t}}=\left(a_{i}^{\hat{t}\_m},a_{i}^{\hat{t}\_mt},s_{i}^{\hat{t}}\right),\;where\;s_{i}^{\hat{t}}\in S^{\hat{t}}$$

The model_artifact_service provides the function to control the model artifact on the cloud, the model_artifact_service can obtain, update, and delete the model artifact in model artifact management. The model_artifact_service also provides the API to let another service obtain the artifact to perform deep learning. The model_artifact_service contains a model artifact controller and artifact switching host. The model artifact controller provides the ability to manage the model artifact and handle the model retrieve request. The artifact switching host can provide the model switching feature. The mode switching is a function that can record which MiniDeep edge device uses the artifact service. If the model artifact has been updated, the next time, the MiniDeep edge device will know the new model available and download it. The training_status_handle_service is placed in the lambda service to serve as a microservice.

All of the above management components and microservices are the core feature of the Train Job Creator. The whole functionality can be access by the AWS console. The MiniDeep edge also provides API to control the Train Job Creator. After all, if the MiniDeep user finishes the model training logic, the model deployer manages and controls the model management.

### 4.3. Training Procedure Details

For the training procedure, the Data Provider (DP) and Train Job Creator (TJC) are important modules to complete the deep learning training process. The detailed procedure of MiniDeep training is shown in Figure 5. This procedure contains a local device. A Data Provider is in the edge device and a Train Job Creator is in the cloud platform. The MiniDeep training procedure uploads the new training data and launches a new training job in this procedure. The detailed procedure is given as:S1.In the beginning, the user sends an ESR_message (*De_id, event_type, event_opt, d_set_name*) to the event_schedule_service and creates an *f_event* on the cloud. The event_schedule_service sets the event by *event_type* and *event_opt*. After setting the event, if the APP has the new training data, the APP will send the training data to the new training cycle. The APP sends a message which is called DP_ins ($De\_id,d\_set\_name,d_{i}^{set}⟶d_{i+d\_num}^{set}$) to the data_collection_service for training data insertion. After receiving *DP_ins*, the data_collection_service indicates the d_set_name, performing StoreUnsync ($d\_set\_name,d_{i}^{set}⟶d_{i+d\_num}^{set}$) to store data in the edge database on the training_dataset_service.S2.After the edge data are stored in S1, the cloud will fetch data when a fetch event occurred, and the fetch event is named *f_event*. When *f_event* occurred, the event_schedule_service calls CloudWatchTriggerEmit (*d_set_name, event_id*) to control the data_synchronize_trigger_service and the data_synchronize_trigger_service to send a DS_fetch_message ( *fetch_id, d_set_name*) so as to fetch the new training data. The training_dataset_service receives the DS_fetch_message and issues an SD_response ($fetch\_id,d_{i}^{set}⟶d_{i+num}^{set},d\_set\_name$) with the new training data.S3.When the APP starts a new training job, the user sends ILTJ_message($De\_id,d\_set\_name,s_{i}^{\hat{t}}$), determining the training script, to the model_instance_launch_service. After the model_instance_launch_ service receives the ITLJ_message, the model_instance_launch_service will obtain dataset $d^{set\_i}$ from d_set_name in the cloud database and obtain the training script $s_{i}^{\hat{t}}$ from the cloud database. After obtaining all elements, the model_instance_launch_service makes a T_JOB object which records training job information and pushes training job information into JOB_QUEUE. The JOB_QUEUE can adjust resource usage to prevent resource usage overloading by setting. When the T_JOB object is popped out, the model_instance_launch_service launches training job by T_JOB_Launching( T_JOB, $s_{i}^{\hat{t}}$, $d^{set\_i}$) and then performs model training on the cloud service provided by AWS.S4.When the model training is finished, the model_instance_launch_service calls MS_artifact ($a^{i}$, d_set_name, $s_{i}^{\hat{t}}$) on the model_artifact_service to store the model artifact in the cloud database. The model_instance_launch_service also calls MTC_event ($a^{i}\_id$, DE_id) to the training_status_ handle_service. The training_status_handle_service obtains MTC_event and knows which APP device needs to be noticed. The training_status_handle_service sends the TJS_publish ($a^{i}\_id$, T_JOB_INFO) message as successful training job information to the APP.

As shown in Figure 5, the model runs S1 (Edge data storing) by sending an ESR_message to perform event schedule registration on TJC. The user can insert new data by sending DP_ins to DP in the edge device, and the edge device stores the new data temporarily until S2 (Cloud data fetching) is executed. When S2 is executed, the TJC sends a DS_fetch_message to DP, and then DP sends back a SD_response. When the APP wants to launch a new training job, the training procedure runs S3 (Performing model training process). In S3, the user sends an ILTJ_message to create the new T_JOB object and then pushes into JOB_QUEUE. The model training will start when the T_JOB object is popped from JOB_QUEUE. After the model training is completed, the training procedure runs S4 (Handling model training result). This step handles the model result storing by performing MS_artifact and sends a notification by sending TJS_publish.

### 4.4. Model Deployer

Figure 6 shows the detailed functions of Model Deployer (MD). Model Deployer on the cloud contains many core services and resource management components. The management components in Model Deployer are the inference script management, inference instance management, and greengrass core management. The inference script management component has the inference script set $S^{\hat{m}}$ uploaded from the local device and stores $S^{\hat{m}}$ to the simple storage service on AWS. The inference script set $S^{\hat{m}}$ has many inference script elements $s_{i}^{\hat{m}}$ in it. One inference script element $s_{i}^{\hat{m}}$ contains an inference script which defines the code of inference. The script meta defines the extra configuration and descriptions of the inference script, such as the model algorithm, model training dataset, etc.

The inference instance management component has several lambda environment definitions, which are deployed to the local device. The lambda environment set $L^{\hat{m}}$ has many lambda environments $l_{i}^{\hat{m}}$. Each lambda environment $l_{i}^{\hat{m}}$ has a lambda instance being used in running specific code in the edge device. env_var is the environment variable setting which is stored in this component and will be applied on lambda instance after lambda instance is deployed in the edge device. The inference resource $r_{i}^{\hat{m}}$ is a resource definition of the lambda instance. The lambda function can achieve the model artifact $a_{i}^{\hat{t}}$ and the inference script $s_{i}^{\hat{m}}$. The inference instance $l_{i}^{\hat{m}}$ uses inference script $s_{i}^{\hat{m}}$ to be a bootstrap role to launch the inference process. $a_{i}^{\hat{m}}$ is the deep learning model with trained weight.

The greengrass core management contains greengrass core set $C^{\hat{m}}$ in which the greengrass core element $C_{i}^{\hat{m}}$ is the whole package which can be deployed to the edge device with many inference instance elements $l_{i}^{\hat{m}},\cdots ,l_{n}^{\hat{m}}$. A greengrass core element has one core component that is a service endpoint of greengrass. Many inference instances in the greengrass core have different inference purposes in the edge device. There also is a D_list that contains many edge device information pieces $d_{i}^{\hat{m}},\cdots ,d_{n}^{\hat{m}}$. Each $d_{i}^{\hat{m}}$ indicates one edge device that the user marked as a tag. Finally, there is a sub_list that contains many elements of subscription information, sub_list uses to manage how the edge device can communicate with the local device by subscribing a pattern style.

The service in model deployer is the core function to finish our deployment job. The inference_package_service in Model Deployer can be called by the outside local application request. The inference_package_service can download model artifact $a_{i}^{\hat{t}}$ from the model_artifact_service of Train Job Creator and package the $a_{i}^{\hat{t}}$ and $s_{i}^{\hat{m}}$ into the inference instance element. The service contains an inference template which is used to create the base inference lambda instance and also uses env_var to assign the environment variable to the inference instance.

The core_packager_service also can handle a local application request that creates the greengrass core and deploys a greengrass core in the edge device. The core packager has an inference instance controller to choose several inference instances $l_{i}^{\hat{m}}$ and package all the choosing inference instances $l_{i}^{\hat{m}}$ into a greengrass core $c_{i}^{\hat{m}}$. Then, the core package handler can handle the request that is responsible for deploying greengrass core $c_{i}^{\hat{m}}$ to inference engine handler in the edge device. The service contains several subscription settings, which is used in the MiniDeep system and also has the device configuration template to set the greengrass device configuration with the uniform format in MiniDeep. We also have a functions definition for the MiniDeep system, which defines a template function creation and assignment in our system.

### 4.5. Inference Engine Handler

The inference engine handler is a core module whose main mission is to host a deep learning inference environment to help the local device access the deep learning computing resource. The inference engine handler contains a greengrass_core_loader_service and openvino_handler_service to achieve the main mission. The overview of the detailed function of the inference engine handler is shown in Figure 7.

The greengrass_core_loader_service has a greengrass core updater to obtain the new greengrass core $c_{i}^{\hat{m}}$ from the greengrass service of AWS. The greengrass core updater loads the greengrass core and unpacks the greengrass core to several lambda inference instances $l_{i}^{\hat{m}}$. Each lambada inference instance contains a model artifact $a_{i}^{\hat{t}}$ that can be put into openvino as an inference model, and an inference script $s_{i}^{\hat{m}}$ can be called as an inference entry point. Each lambda inference instance $l_{i}^{\hat{m}}$ contains the lambda instance, env_var, and all of the basic elements are a host on the local lambda runtime. The greengrass_core_loader_service also has an inference function selector, which can control what lambda inference instance is enabled or not, and the local application can decide each lambda inference instance process.

The openvino_handler_service hosts an openvino inference engine and an openvino model optimizer. The openvino model optimizer is an openvino core function which can parse the deep learning model generated from the deep learning platform, such as tensorflow, Mxnet, caffe, etc. The model optimizer converts the deep learning model to the IR (Intermediate Representation) file. The IR file consists of the *.bin file as the model weight and *.xml as the deep learning model architecture. The inference engine reads the IR files, and it hosts a deep learning computing acceleration environment for the IR model. In our openvino_handler_service, MiniDeep has an inference endpoint to handle the outside HTTP request for inference and a model loader to load the deep learning model to the model optimizer to obtain the IR file. The openvino host function can read the inference instance resource and bring the inference instance resource to the model loader to obtain the IR and put the IR to the inference endpoint. The local application can send HTTP to obtain the request of the inference and then obtain the response of the inference.

### 4.6. Inference Procedure Details

There are two main microservices, the model deployer and the inference engine handler. Figure 8 shows the procedure of how two microservices work together to perform the inference procedure.

S1.The user sends an IIC_message (De_id, $a_{id}^{i}$, $s_{i}^{\hat{m}}$) to the inference_packager_service to instantiate the lambda instance $L_{i}^{\hat{m}}$. The inference_packager_service receives the IIC_message and tries to obtain data by sending the MA_get_message ($a_{id}^{i}$) to the model_artifact_service. The model_artifact_service receives the MA_get_message and responds with an MA_res_message ($a_{i}$) which contains the designated model artifact $a_{i}$. After the inference_packager_service obtains the model artifact $a_{i}$, the inference_packager_service starts LII_start ($a_{i}$, $s_{i}^{\hat{m}}$). After finishing LII_start, the inference_packager_ service produces an L_ins, sends an IIC_ok_message (L_ins_id) and then finishes the setup of the inference instance.S2.The user creates a package with multiple L_ins. The APP sends a message IIP_message (De_id, [L_ins_id⋯ ], $ENV_{V}AR$) to the core_packager_service. The core_packager_service receives an IIP_message and starts the packaging process. The core_packager_service first retrieves many L_ins from the inference_packager_service by performing the II_retrieve ([L_ins_id]), and the inference_packager_service provides many L_ins to the core_packager_service by performing II_retrieve_res ([L_ins⋯]). The core_packager_service obtains the list of L_ins, packages L_ins as a L_ins_set and creates a GGCore instance. After that, the core_packager_service sends back a message which is called IIP_ok_message (GGCore_id) with the green grass core ID.S3.The user deploys the inference-able model to the edge device. The APP sends a GGC_deploy_ message (De_id, GGCore_id) to the greengrass_core_loader_service to request a deployment. The greengrass_core_loader_service receives a request message called GGC_deploy_message and sends a GGC_get_message (GGCore_id) to the core_packager_service. Then, the core_packager_ service responds with a GGC_get_res (GGCore) which contains the GGCore instance. The greengrass_core_loader_service loads the GGCore and then passes GGCore into Model Optimizer to obtain the IR file. The IR file is passed to the openvino_handler_service by performing GGC_serve (IR) to host an inference serving endpoint.S4.In this step, the user wants to send an inference response and train new data if the data are the new ground true data. The APP sends an Infer_req ($De_{id}$, $x^{i}$, GGCore_id, L_ins_id) to the openvino_handler_service. The openvino_handler_service handles an inference request and passes $x^{i}$ into the L_ins in the GGCore and then obtains an inference result $y^{i}$. After producing inference results in $y^{i}$, the openvino_handler_service sends back a Infer_res ($y^{i}$) to the APP. The APP can realize that the result is accurate or not by human selection. If the human selection is not good, the APP can choose to send a DP_ins to send new data.

As shown in Figure 8, the user executes S1 (inference instance setup) by sending a IIC_message to perform inference instance creation on MD (Model Deployer). MD obtains artifact $a_{i}$ by sending the MA_get_message to TJC in the cloud, and performs LII_start to instantiate the lambda instance L_ins. In step S2 (Inference instance packaging), the user sends an IIP_message to select many L_ins to package L_ins together and then creates a GGCore instance in MD. In step S3 (Inference Instance Deployment), the user sends a GGC_deploy_message to assign GGCore into IEH (Inference Engine Handler). The IEH loads the GGCore and converts GGCore into OpenVino-known file format, IR file, and hosts an inference request endpoint by performing GGC_serve. In step S4 (Inference Request Handling), the user sends Infer_req to the IEH inference endpoint, and the IEH infers the data by performing DNN computation on OpenVino. The IEH sends Infer_res back to APP with inference result $y_{i}$. After producing inference results in $y^{i}$, the openvino_handler_service sends back an Infer_res ($y^{i}$) to the APP. The APP can realize that the result is accurate or not by human selection. If the human selection is not good, the APP can choose to send a DP_ins.

## 5. AI-QSR KIOSK Application Software Architecture

This section introduces the AI-QSR system architecture in Section 5.1. An LSTM-based recommendation system design for the AI-QSR system is described in Section 5.2. The MiniDeep platform constructed in the AI-QSR system is described in Section 5.3.

### 5.1. AI-QSR System Architecture

In this paper, the AI-QSR (Artificial Intelligence-Quick Service Restaurant) KIOSK (interactive kiosk) is proposed to demonstrate the MiniDeep platform result. The AI-QSR system architecture is shown in Figure 9. There are two environments in the system: one is a quick service restaurant environment, and the other is the MiniDeep cloud service environment. The quick service restaurant environment has one KIOSK instance with a camera device, and the KIOSK instance contains a quick service restaurant application in this KIOSK. There is a MINI-PC with a MiniDeep platform inside the KIOSK, and the MINI-PC uses a network cable to connect with the KIOSK. The role of the MINI-PC is an edge device and provides a deep learning inference resource to quick service restaurant application. To provide a real-time experiment of the inference process, the MINI-PC uses an Intel Movidius Neural Compute Stick to speed up the computation time of the neural network operations. The MINI-PC also runs a recommendation system which is written with the python language and hosts as a RESTful API endpoint to handle the HTTP request from the quick service restaurant application. On the cloud environment, MiniDeep uses two main services to serve the model training and model deployment. The model training uses the sagemaker service to manage the training environment. The model deployment uses the greengrass service to manage the model packaging and configuration deployment.

In the AI-QSR system architecture, the quick service restaurant application served in KIOSK can provide a friendly user interface to show the menu, food, suit, and applications. The AI-QSR has a feature that can recommend the most likely purchased food to the user. When the KIOSK user clicks the element of application such as food, menu, or suit, the application will check the condition and decide to make a recommendation to the user. When the application needs a recommendation, the application will send a request to the recommendation system on the MINI-PC, and the recommendation system collects the user face images taken from the camera and the click event as a flow-sequence to request the inference result from the MiniDeep platform. The Inference Engine Handler accepts the inference request and computes an inference result for the recommendation system. This recommendation system sends back the inference result to the application. Then, the application shows the recommendation result.

### 5.2. LSTM-Based Recommendation System Design for AI-QSR

The recommendation problem in the scenario of the AI-QSR system is regarded as the QSR-FS problem, which is described in Section 3.4, and the formulation of the QSR-FS problem is described in Section 3.5. The problem of the QSR-FS is to achieve the highest hit purchase accuracy. An LSTM-based recommendation system is designed for the QSR-FS problem of the AI-QSR system. The system is shown in Figure 10. There are three system structures that perform the transfer learning technology. The three system structures of the AI-QSR are the pre-trained age–gender estimator, the purposed LSTM of the QSR-FS problem, and the food classified DNN.

Except for the deep learning model, the QSR-FS problem also needs two function blocks and three function triggers to work together. The two function blocks are the recommend controller and the food item aggregator. The recommend controller controls the trigger event and operates the other function block. The food item aggregator collects the food item candidate from the LSTM of the QSR-FS problem. If the recommend controller sends a control message to the food item aggregator, the food item aggregator will send its output to calculate the recommendation result. The three function triggers are the event entries that every event entry can use to trigger the recommend controller to start a recommendation candidate selection and food item recommendation. The three function triggers are the popup trigger, the wait trigger, and the browsing trigger. The popup trigger comes from the popup event in the AI-QSR. The wait trigger comes from the user’s waiting time exceeding the timeout. The browsing trigger comes from the user’s clicking count that without purchasing anything.

For the QSR-FS problem, our training data are represented as $x_{i}=\{d_{i}^{a},d_{i}^{g},d_{i}^{c},(f_{1},\cdots ,f_{j},\cdots ,f_{s})\}$, and the labeled data are represented as $y_{i}=\left\{f_{j}^{p},f_{j}^{ca},f_{j}^{m},f_{j}^{co}\right\}$. To obtain the recommend food result, each element in $x_{i}$ and $y_{i}$ needs to be provided. In our system, the pre-trained age–gender estimator can generate the $d_{i}^{a}$ and $d_{i}^{g}$ information from the user’s face image information. The pre-trained age–gender estimator is a Wide Residual Network with the age and gender recognition functionality, which is proposed by Yusuke Uchida (yu4u) on github [27]. For the transfer learning in this scenario, QSR-FS uses a pre-trained model to obtain the middle feature and then uses this middle feature to perform the behind training. The age–gender estimator has an input layer with (64×64) input shape and then connects with three wide residual layers that use a residual model design to solve the gradient descent problem. The end of the wide residual layer connects with an average pooling. The average pooling connects with two fully connective layers for different task purposes. This technology is called the multi-task learning. Two fully connective layers share the whole network’s feature information so that the model can learn more general features to solve the different tasks. One fully connective output $d_{i}^{a}$ represents the user age, and the other fully connective output $d_{i}^{g}$ represents the user gender. The two pieces of information will combine with the food item aggregator result.

The purposed LSTM-based scheme of the QSR-FS problem is an LSTM which parses the QSR-FS click event sequence. This sequence is called the Flow Sequence (FS). The Flow Sequence is a serial of the click events. The user’s clicking of each QSR element on the AI-QSR user interface will be recorded. The click event is converted to the AI-QSR element character. All the AI-QSR element characters are combined as a flow sequence. Each flow sequence contains very lengthy AI-QSR element characters which use one-hit-encoding to represent it. The flow sequence $X=(x_{1},\cdots ,x_{n})$ is an input of LSTM which is sent to LSTM (many-to-one) to compute the time-series flow sequence data and obtain the label $y_{i}$. Then, label $y_{i}$ is fed into the two fully connective layers: one is the buy-or-not fully connective layer, and another is the food-item fully connective layer. The buy-or-not fully connective layer produces one dimension output $f_{i}^{buy}$, and the food-item connective layer produces **food_item_num**. In summary, the detailed procedure of obtaining the recommendation food result with the MiniDeep platform is given below:S1.The LSTM of the QSR-FS problem can compute the $f_{i}^{buy}$ and $f_{i}^{item}$ by feeding the flow sequence FS. In our problem, the AI-QSR needs to collect different Flow Sequences FS to find many food item candidates F_item_c, so that the AI-QSR computes *N* flow sequences (FS) as a Flow Sequence Set (FSS) and collects FSS at the FI_aggregator (food item aggregator).S2.The FI_aggregator collects the food item candidates F_item_c from which the user may buy the food item by checking $f_{i}^{buy}$. The candidate F_item_c that the user may buy can be combined as F_List (Food List). The F_List is combined and flattened to the high dimension vector. The recommendation system continually collects the user Flow Sequence FS and maintains the F_List to hit the recommendation result.S3.When the user hits the T_condition (trigger condition) defined on the AI-QSR application, the AI-QSR application sends an RL_message (recommendation launch message) to R_controller (recommend controller), and the R_controller manages the FI_aggregator and pops out the F_List result.S4.The camera takes a picture of the user and passes the user’s face image to the pre-trained age–gender estimator to obtain $d_{i}^{g}$ and $d_{i}^{a}$. The FI_aggregator sends an F_list and combines F_list with $d_{i}^{g}$ and $d_{i}^{a}$ to a high dimension vector for a food-classified DNN. The food-classified DNN parses the high-dimension vector and passes the vectors with shape 512 and shape 64 to two fully connective layers.S5.At the end of the fully connective layer, the output result is called RF_Result (recommend food result). The RF_Result is a one-hot-encoding vector in which only the suggested food will be 1 and the other output will be 0.

The AI-QSR in S1 converts the Flow Sequence to the food items candidates. In S2, the AI-QSR performs food items aggregation to combine food item candidates together. In S3, AI-QSR launches a recommendation when the user hits the T_condition. In S4, the age–gender detection is going to find the user’s age and gender. Finally, in S5, the AI-QSR calculates the recommendation result by the result of the last fully connective layer.

### 5.3. AI-QSR Recommendation System Implemented by MiniDeep Platform

In the paper, the MiniDeep platform is used to implement the LSTM solution of the QSR-FS problem in the AI-QSR application and is shown in Figure 11. There are three modules in the recommendation system. One is a local hardware devices module, another is the AI-QSR application, and the other is the MiniPc core microservices. With the MiniDeep platform, there is a model deployer that makes a greengrass core which contains a pre-trained age–gender estimator, the LSTM solution of the QSR-FS problem, and a food-classified DNN. The greengrass core of the AI-QSR is deployed to the inference engine handler of the edge device and the greengrass_core_loader_service updates the openvino runtime to host three models. The local KIOSK has two hardware modules, the camera module and the screen module. Every time the user clicks on the KIOSK, the screen module will send a Flow Sequence (FS) to the purposed QSR-FS problem in the inference engine handler, and the LSTM output is stored in the food item aggregator (FI_aggregator). When the user starts using the KIOSK, the user may trigger the popup trigger, the wait trigger, and the browsing trigger. If the user hits the trigger, the trigger sends a trigger_hit_event to the Recommend Controller (RC). RC starts a food-classified recommendation by sending the food_list_message in the FL_aggregator to the inference engine handler, and it combines with the age–gender vector. The age–gender vector comes from the output of the pre-trained age–gender estimator such that the KIOSK camera module takes a photo on the user’s face and send the user’s face image to the pre-trained age–gender estimator to obtain an age–gender vector. After combining the food_list_message and age–gender vector into food-classified feature (FC_feature), the FC_feature is sent to the food-classified DNN and produces the output of the recommending food result, which is the top food the user wants to buy.

## 6. Experimental Result

The experimental environment of the AI-QSR application and the performance analysis of the LSTM-based solution of the QSR-FS problem are shown in this section.

### 6.1. Experimental Environment

There is a big touch-screen KIOSK with a camera in the experimental environment. KIOSK uses a LIVA X Mini PC as the host PC with Intel Bay Trail-M/Bay Trail-I SOC 2.25 GHz and 2GB DDR3 RAM. The AI-QSR front-end interface is the host in LIVA X Mini PC. A MiniDeep platform uses a Liva Q Mini PC as the edge PC with Intel Apollo Lake Celeron N3350 SOC and 4GB/32GB eMMC. The edge PC is connected with the host PC with a network cable, and the edge PC uses LAN to communicate with the host PC. The MiniDeep platform also has the Ethernet capability to access AWS services. An Intel Movidius neural compute stick is plugged in the edge PC so that the MiniDeep platform can perform high-speed acceleration for inference. Figure 12 shows the AI-QSR user interface.

The training data are collected from the Computex 2019 exhibition in Taipei. In total, 623 users’ click data and face data are collected during the 5 days. The training data are collected through the AI-QSR user interface (UI) in the demo mode. Here, 80% of the collected data is used for training and 20% of the collected data is used for testing. After the Computex exhibition, the training result is stored on AWS cloud storage, and then, our team trains the LSTM-based model of the QSR-FS problem with the new training data on the sagemaker cloud service. After completed training, the trained model is deployed to the edge PC of AI-QSR KIOSK.

When the AI-QSR UI uses AI mode, the AI-QSR recommends food order with the inference result of the LSTM-based solution of the QSR-FS problem. In Section 6.2, many performance analyses of the deep learning classification problem are discussed. Our approach shows that the accuracy and system performance is higher than the rule-based recommendation system which is the baseline to evaluate the performance. Furthermore, our AI-QSR KIOSK system has the following recommendation options: the popup-triggered recommendation, the idle-triggered recommendation, and the browsing-triggered recommendation, which are shown in Figure 13 and Figure 14.

### 6.2. Performance Analysis

The performance analysis is presented in this section. There are three recommend triggers: the popup trigger, the idle trigger, and the browsing trigger. Figure 15 shows the time slot definitions of the three triggers. The three types of time proportion ($\hat{p}$, $\hat{w}$, $\hat{b}$) are set as (0.8, 0.1, 0.1), (0.1, 0.8, 0.1), and (0.1, 0.1, 0.8) so as to demonstrate the impact of the recommendations by the three different triggers, where $\hat{p}$ represents the popup proportion, $\hat{w}$ represents the idle proportion, and $\hat{b}$ represents the browsing proportion. The popup-triggered recommendation appears after identifying the gender and age of the users. The browsing-triggered recommendation appears during the browsing of the webpages. The idle-triggered recommendation appears when the user stays (or idles) on a webpage for a certain period of time.

The experiment compares the rule-based scheme with the proposed LSTM-based scheme and uses some metrics to evaluate the performance of the two schemes. The performance metrics are defined as follows.

Purchase hit accuracy: The matching possibility of the food recommending by the scheme and users’ purchases.Categorical cross-entropy: By calculating the size of the loss function, categorical cross-entropy is the main basis in the learning process and an important criterion for judging the merits of the algorithm after learning.Precision: Precision is the ratio of all “correctly retrieved results (True Positive)” to all “actually retrieved (True Positive + False Positive)”.Recall: Recall is the ratio of all “correctly retrieved results (True Positive)” to all “results that should be retrieved (True Positive + False Negative)”.F1 score: F1 score is the weighted harmonic average of Precision and Recall. When the F1 score is high, it can be proved that the test scheme is effective.

#### 6.2.1. Purchase Hit Accuracy

The purchase hit accuracy is defined as follows.
(19)$$\begin{array}{c} A\approx \underset{\theta^{*}}{\arg\max}\left\{e^{-\underset{\theta }{\arg\min}\left\{\frac{1}{N}\cdot \sum_{i=1}^{n}\cdot L\left(y_{i},f\left(x_{i},\theta \right)\right)+\lambda \Phi \left(\theta \right)\right\}}\right\}\\ \;,\;subject\;to\left\{\begin{array}{c} X=(x_{0},\cdots ,x_{i},\cdots ,x_{n})\\ Y=(y_{0},\cdots ,y_{i},\cdots ,y_{n}) \end{array}\right.\\ \; \end{array}$$

Figure 16 shows that when $\theta^{*}$ becomes greater during the training process, the higher the purchase hit accuracy becomes. The purchase hit accuracy of the LSTM-based scheme is higher than that of the rule-based scheme when the epoch is greater than 1800. Since we use the users’ age, gender, and the click event sequences to train our AI-QSR recommendation system, the proposed LSTM-based scheme can make a more accurate prediction (or recommendation), and thus, the proposed LSTM-based scheme can achieve higher purchase hit accuracy.

Among the three types of time proportion, the browsing-triggered recommendation recommends products from the contents the users repeatedly search and browse, and thus, it is the most accurate. The click rate of the products recommended by the idle-triggered recommendation is the lowest because the recommendation is made through too little information of the users. The accuracy of the popup-triggered recommendation is worse than that of the browsing-triggered recommendation because the identification accuracy of age and gender is not accurate enough or because of the regional cultural differences.

#### 6.2.2. Categorical Cross-Entropy

The cross-entropy loss function is shown as follows, where *n* is the total number of the training data, $x_{j}$ is the *j*-th training input, $y_{j}$ is the *j*-th desired output, and $a_{j}$ is the *j*-th output of the neural network.
(20)$$\begin{array}{c} C=-\frac{1}{n}\sum_{x}\sum_{j}\left[y_{j}\ln a_{j}+\left(1-y_{j}\right)\ln\left(1-a_{j}\right)\right] \end{array}$$

The categorical cross-entropy loss is the sum of the cross-entropy loss of each category. The smaller the categorical cross-entropy loss is, the training result is more likely to fit the true world data. Figure 17 shows that the longer the training process is, the lower the categorical cross-entropy loss becomes. The loss of the LSTM-based scheme is lower than that of the rule-based scheme when the epoch is greater than 1800. Since the proposed LSTM-based scheme can make a more accurate prediction (or recommendation), the proposed LSTM-based scheme can achieve lower categorical cross-entropy loss.

Among the three types of time proportion, since the browsing-dominated time proportion is the most accurate, followed by the popup and idle-dominated time proportion, the categorical cross-entropy loss of the browsing-dominated time proportion is the lowest, which is followed by the popup and idle-dominated time proportion.

#### 6.2.3. Precision

The precision of the confusion matrix is shown as follows.
(21)$$\begin{array}{c} precision=\frac{TP}{TP+FP} \end{array}$$

Figure 18 shows that the longer the training process is, the higher the precision becomes. The precision of the LSTM-based scheme is higher than that of the rule-based scheme when the epoch is greater than 1800. Since the proposed LSTM-based scheme can make a more accurate prediction (or recommendation), the proposed LSTM-based scheme can achieve higher precision.

#### 6.2.4. Recall

The recall of the confusion matrix is shown as follows.
(22)$$\begin{array}{c} recall=\frac{TP}{TP+FN} \end{array}$$

Figure 19 shows that when the training process is longer, the higher the recall becomes. The recall of the LSTM-based scheme is higher than that of the rule-based scheme when the epoch is greater than 1500. Since the proposed LSTM-based scheme can make a more accurate prediction (or recommendation), the proposed LSTM-based scheme can achieve higher recall.

#### 6.2.5. F1 Score

The F1 score of the confusion matrix is shown as follows.
(23)$$\begin{array}{c} F1=\frac{2*TP}{2*TP+FP+FN} \end{array}$$

Figure 20 shows that when the training process is longer, the higher the F1 score becomes. The F1 score of the LSTM-based scheme is higher than that of the rule-based scheme when the epoch is greater than 1500. Since the proposed LSTM-based scheme can make a more accurate prediction (or recommendation), the proposed LSTM-based scheme can achieve a higher F1 score.

#### 6.2.6. Iterations

The training iterations is a hyperparameter. Figure 21 shows that as the iteration increases, the purchase hit accuracy, recall, and F1 score also increases, but the categorical cross-entropy loss decreases. Overall, the proposed LSTM-based scheme performs better than the rule-based scheme.

## 7. Conclusions

In this paper, we have proposed a new AI-Edge platform called MiniDeep. The MiniDeep platform provides developers with a whole deep learning development environment to set up their deep learning life cycle processes, such as model training, model evaluation, model deployment, model inference, ground truth collecting, data pre-processing, and training data management. To the best of our knowledge, such a whole deep learning development environment has not been proposed before. We build a Deep Learning-Based MINI-PC and an AI-QSR KIOSK system with food recommendations on the MiniDeep platform. The recommendation system uses an LSTM-based scheme to solve the QSR-FS problem. The experiment results show that the proposed LSTM-based scheme performs better than the rule-based scheme in terms of purchase hit accuracy, categorical cross-entropy, precision, recall, and F1 score. These results demonstrate the effectiveness of the proposed AI-Edge platform. In the future, we plan to improve the generality and usability of the MiniDeep platform so that more AI-based applications can be built on the MiniDeep platform.

## Figures and Tables

**Figure 1 sensors-22-05975-f001:**
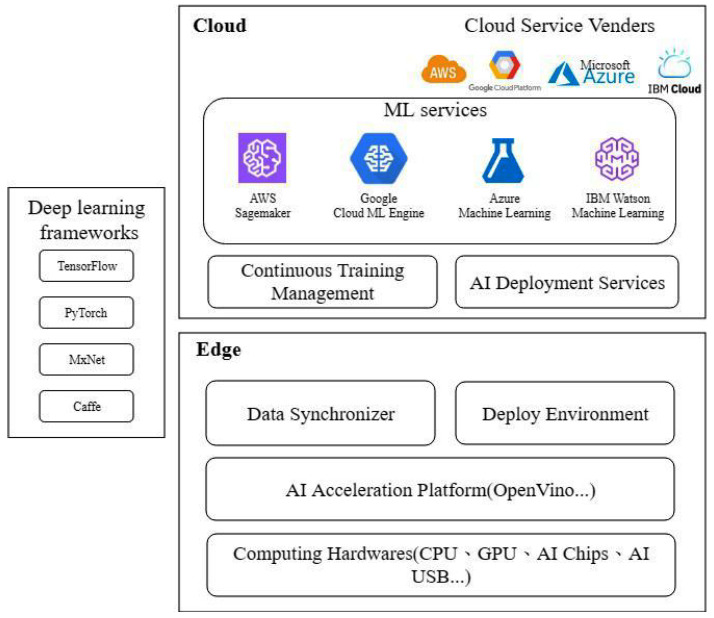
The system architecture of the edge-cloud model.

**Figure 2 sensors-22-05975-f002:**
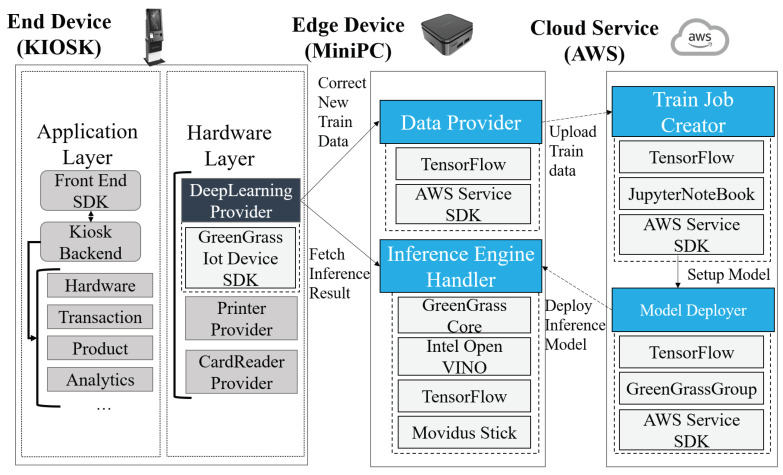
The system archicture of the MiniDeep platform.

**Figure 3 sensors-22-05975-f003:**
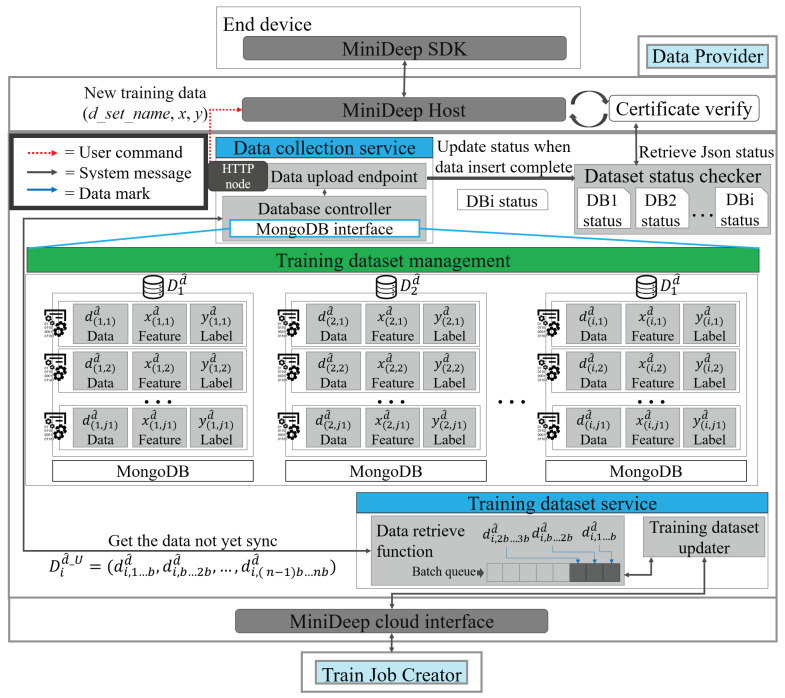
The detailed functions of the Data Provider (DP).

**Figure 4 sensors-22-05975-f004:**
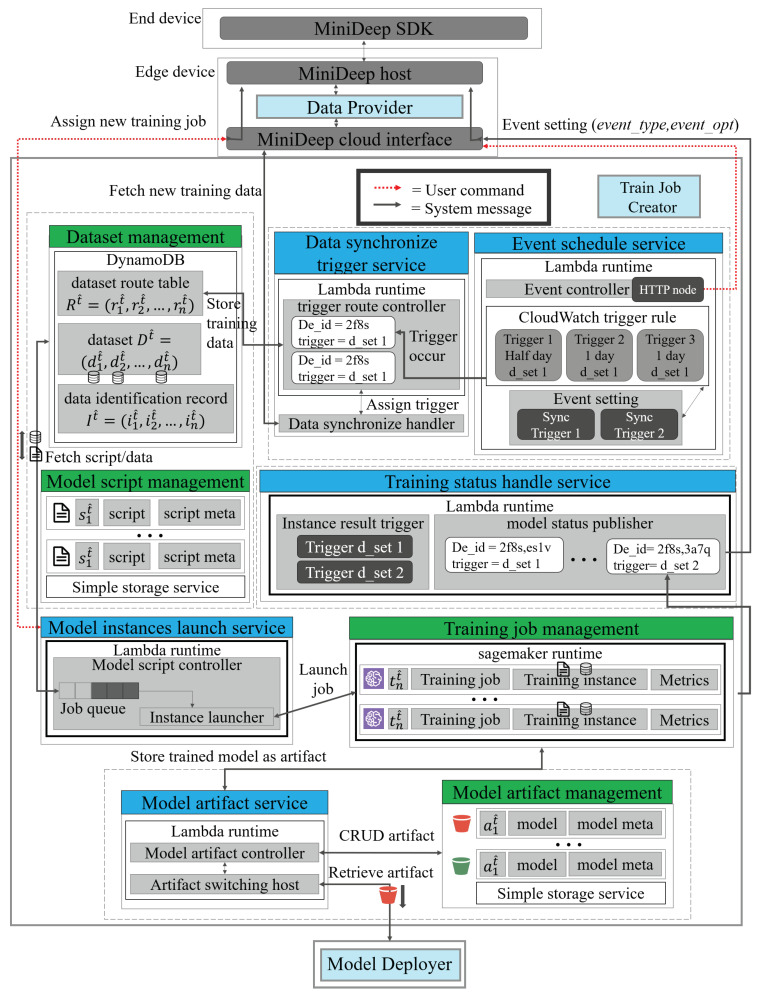
The detailed functions of the Train Job Creator.

**Figure 5 sensors-22-05975-f005:**
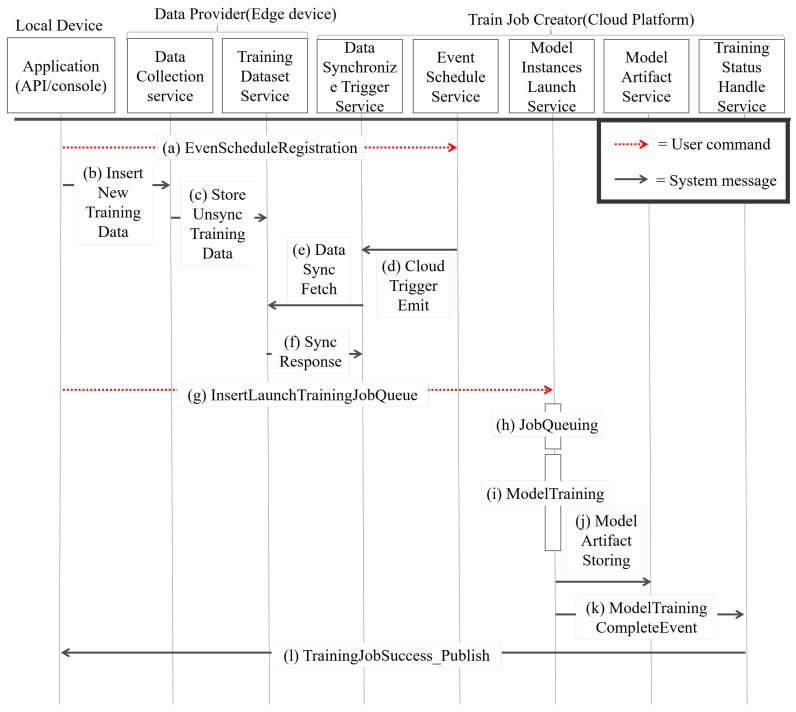
The training procedure on the MiniDeep platform.

**Figure 6 sensors-22-05975-f006:**
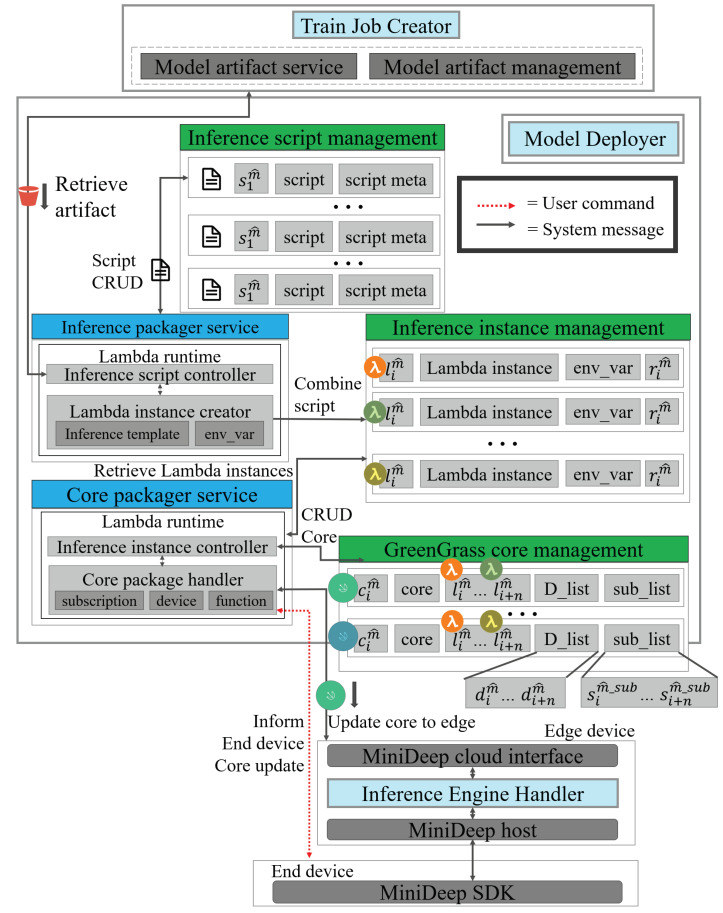
The detailed functions of Model Deployer.

**Figure 7 sensors-22-05975-f007:**
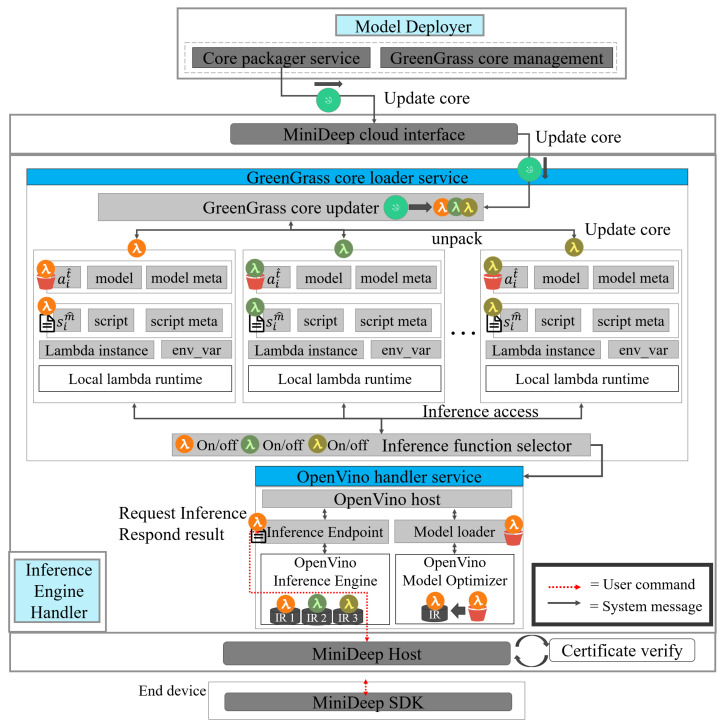
The detailed function of the inference engine handler.

**Figure 8 sensors-22-05975-f008:**
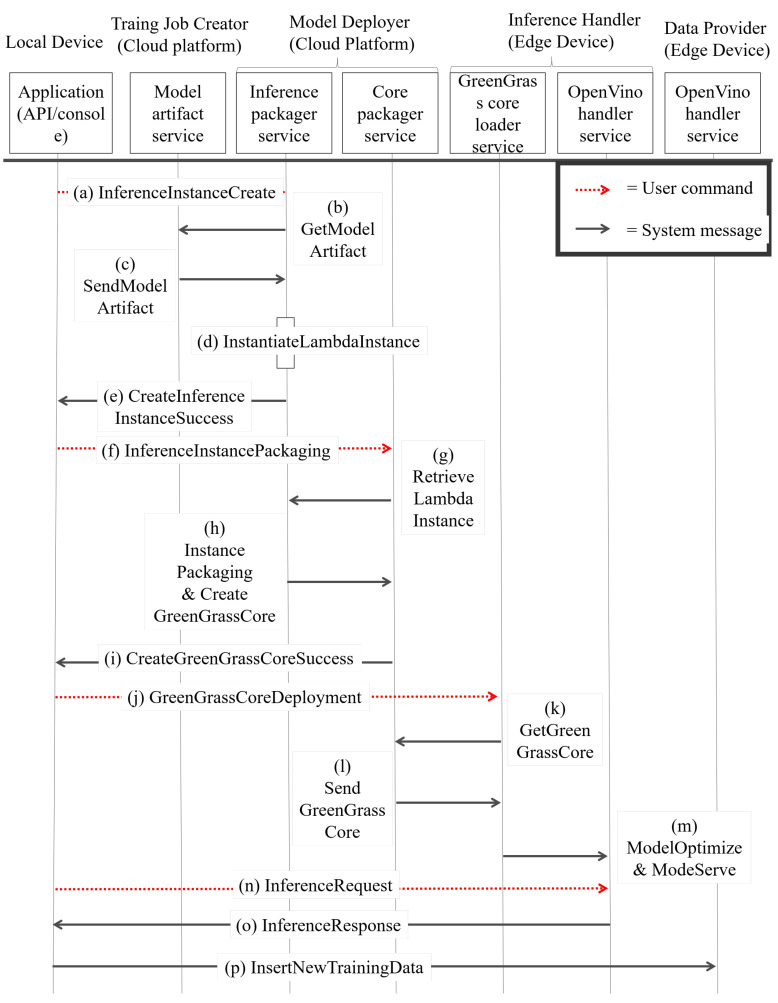
The inference procedure on the MiniDeep platform.

**Figure 9 sensors-22-05975-f009:**
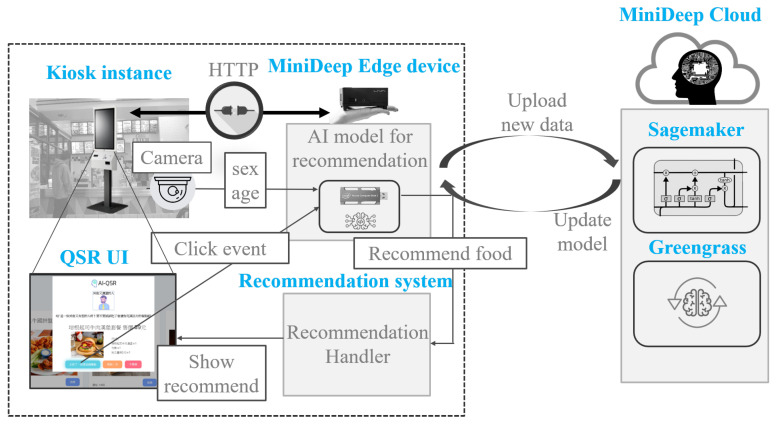
The AI-QSR system architecture.

**Figure 10 sensors-22-05975-f010:**
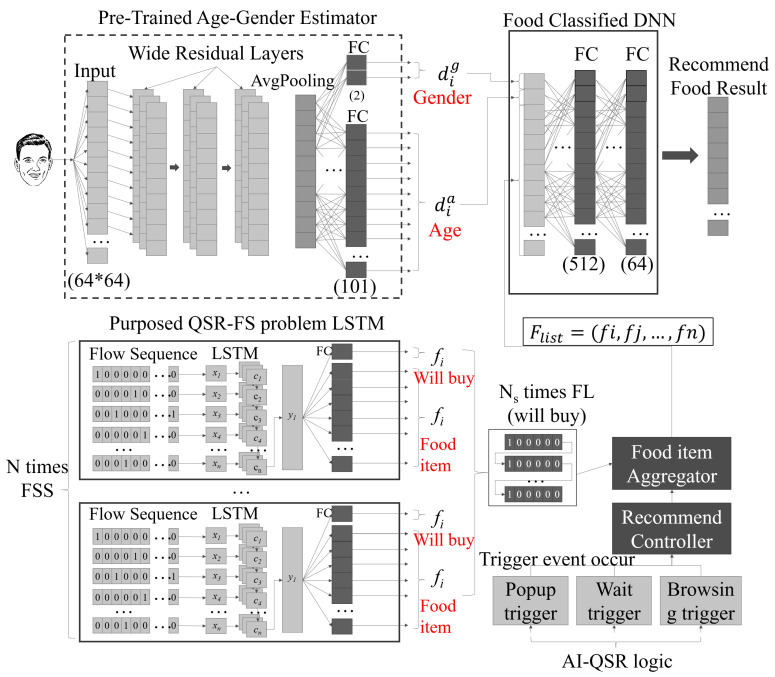
The AI-QSR system.

**Figure 11 sensors-22-05975-f011:**
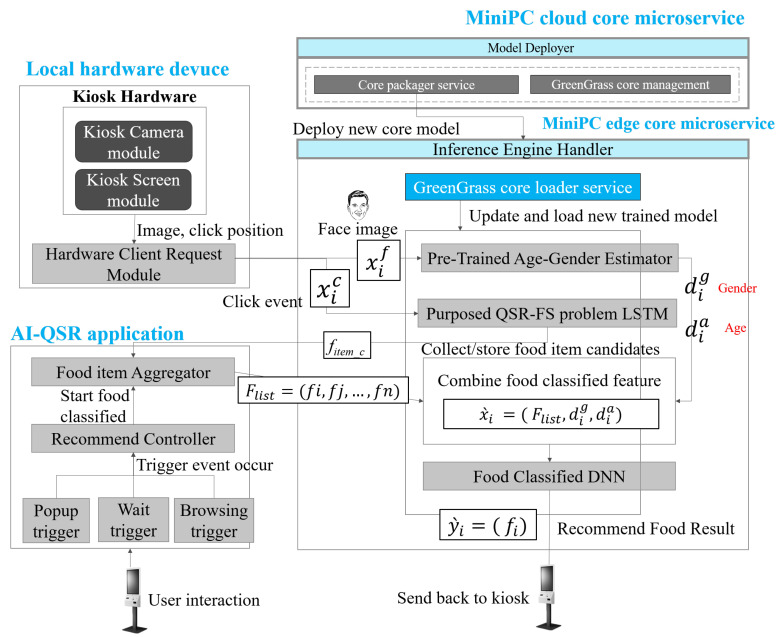
The AI-QSR recommendation system.

**Figure 12 sensors-22-05975-f012:**
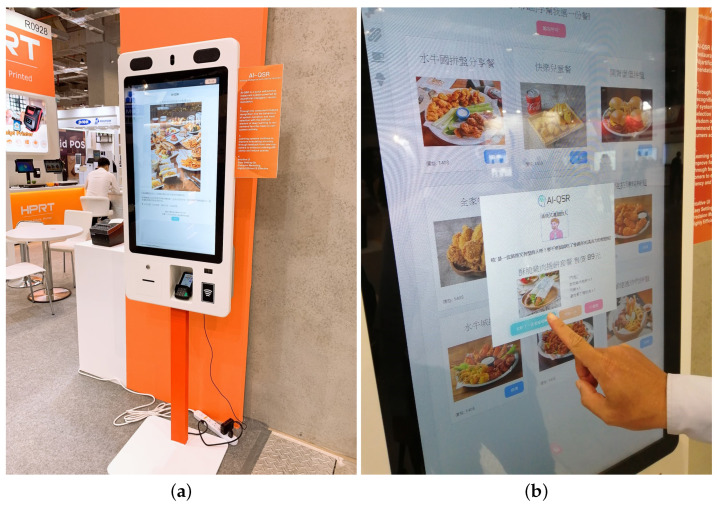
The AI-QSR user interface: (**a**) The AI-QSR KIOSK system. (**b**) The using of the AI-QSR KIOSK system.

**Figure 13 sensors-22-05975-f013:**
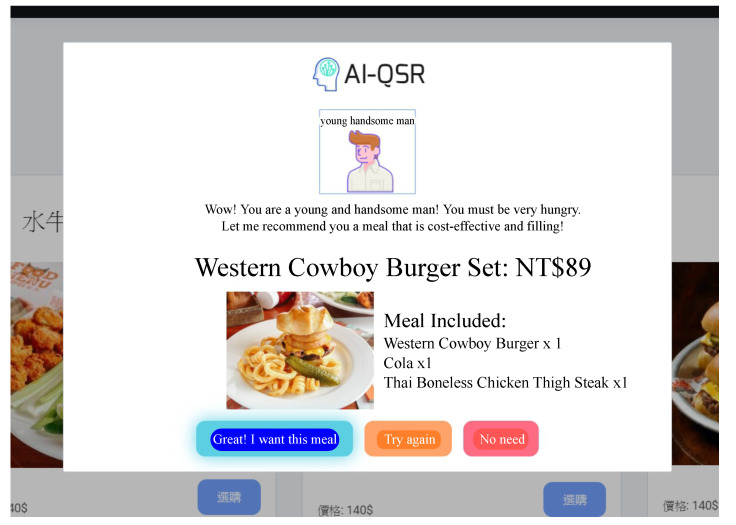
The UI of the popup-triggered recommendation.

**Figure 14 sensors-22-05975-f014:**
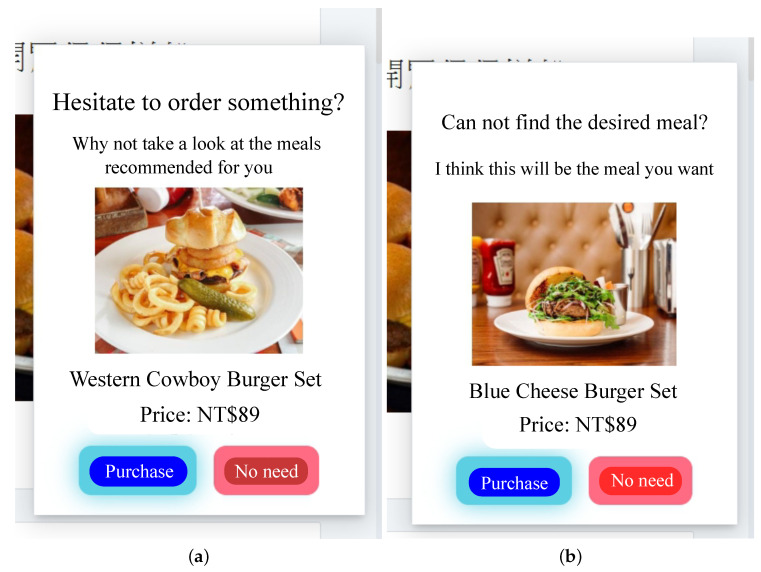
Different types of notifications on the AI-QSR UI. (**a**) The UI of the idle-triggered recommendation. (**b**) The UI of the browsing-triggered recommendation.

**Figure 15 sensors-22-05975-f015:**
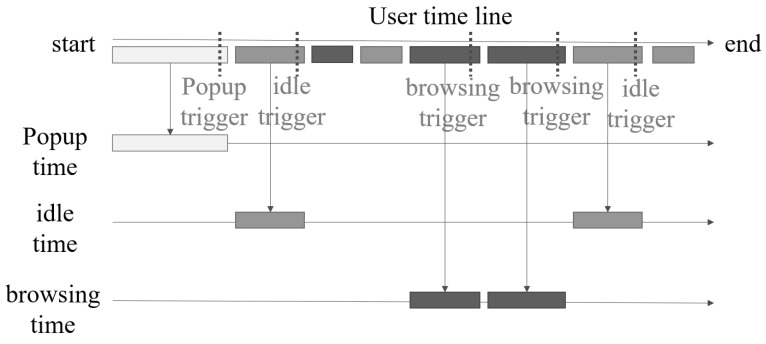
Overview of the AI-QSR time slot definitions.

**Figure 16 sensors-22-05975-f016:**
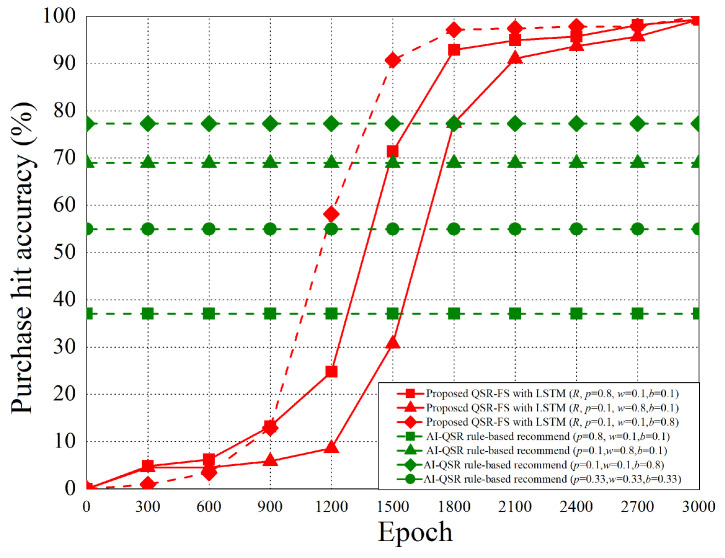
Purchase hit accuracy vs. per epoch.

**Figure 17 sensors-22-05975-f017:**
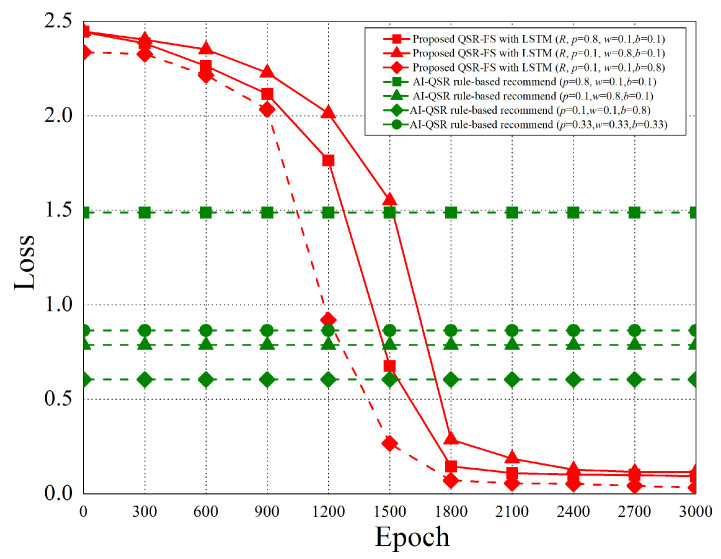
Categorical cross-entropy loss vs. per epoch.

**Figure 18 sensors-22-05975-f018:**
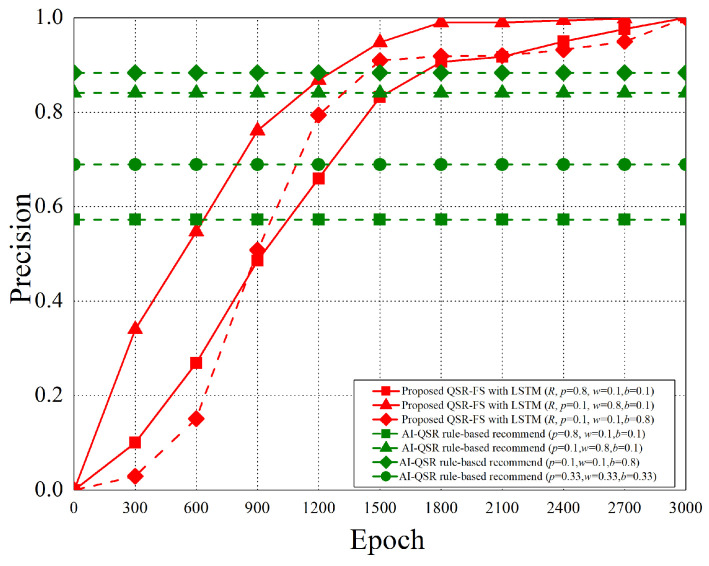
Precision vs. per epoch.

**Figure 19 sensors-22-05975-f019:**
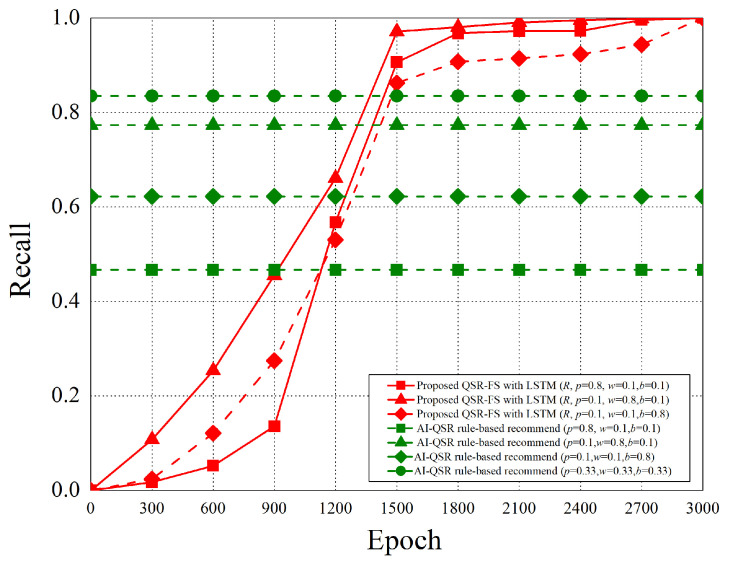
Recall vs. per epoch.

**Figure 20 sensors-22-05975-f020:**
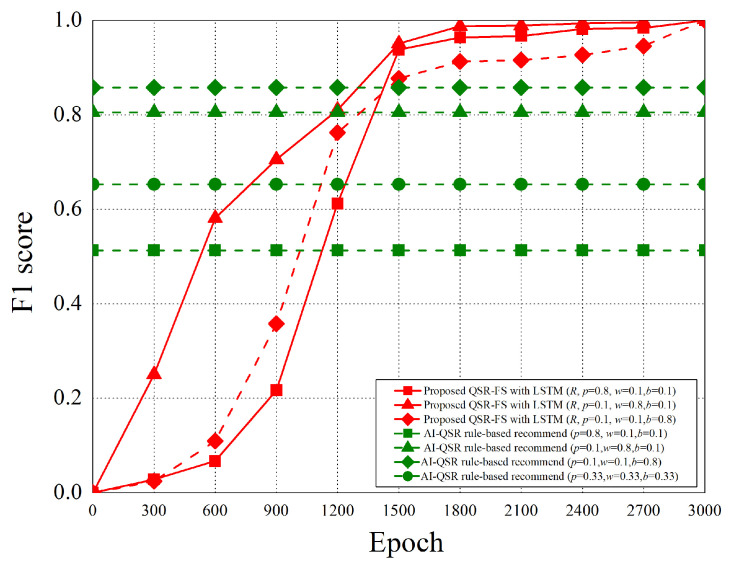
F1 score vs. per epoch.

**Figure 21 sensors-22-05975-f021:**
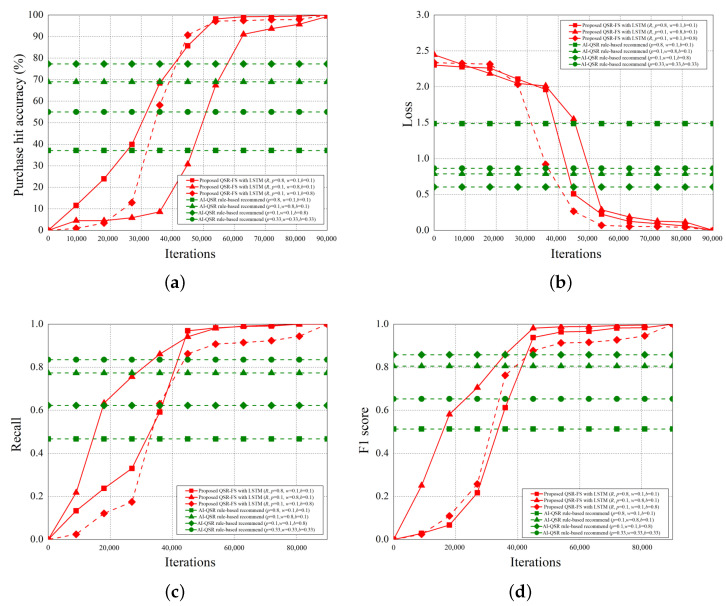
The comparison of iteration between different analysis metrics. (**a**) Purchase hit accuracy vs. per iteration. (**b**) Categorical cross-entropy loss vs. per iteration. (**c**) Recall vs. per iteration. (**d**) F1 score vs. per iteration.

## Data Availability

Not applicable.

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
