# Peer review of "MiniDeep: A Standalone AI-Edge Platform with a Deep Learning-Based MINI-PC and AI-QSR System"

_sensors, 2022, doi:10.3390/s22165975_

Round 1

Reviewer 1 Report

This paper proposed a new AI-Edge platform, and build a deep learning-based MINI-PC and an AI-QSR KIOSK system with food recommendation on the platform. The experiment environment is described and the performance analysis for AI-QSR recommendation system has demonstrated that the LSTM-based scheme performs better than the rule-based scheme. There are some comments as followings.

 1. Some notations are used without definition. Please define every notation clearly.

2. There are still some wording and grammar errors in this paper. Please proofread this paper carefully.

3. Many readers cannot read Chinese. Please translate the texts on Figures 13 and 14 into English.

4. The authors do not explain why the three types of time proportion (popup proportion, idle proportion, browsing proportion) are set as (0.8, 0.1, 0.1), (0.1, 0.8, 0.1), and (0.1, 0.1, 0.8). What are the impacts of the time proportion to the performance?

5. The authors do not describe the future works of this paper.

Author Response

We have revised the paper carefully according the reviewers’ comments. Each of the comment has been responded carefully. We have proofread the paper and revised many sentences and typos in this paper. Hence, we cannot highlight every minor revision sentence and words. Otherwise, more than a third of the sentences in this paper will be highlighted. Therefore, only the major revision parts have been highlighted.

Reviewer 1:

This paper proposed a new AI-Edge platform, and build a deep learning-based MINI-PC and an AI-QSR KIOSK system with food recommendation on the platform. The experiment environment is described and the performance analysis for AI-QSR recommendation system has demonstrated that the LSTM-based scheme performs better than the rule-based scheme. There are some comments as followings.

Some notations are used without definition. Please define every notation clearly.

Answer:

We have defined all the notations clearly in this paper.

There are still some wording and grammar errors in this paper. Please proofread this paper carefully.

Answer:

We have proofread the paper carefully so as to avoid any wording and grammar errors.

Many readers cannot read Chinese. Please translate the texts on Figures 13 and 14 into English.

Answer:

We agree with you and have translated the texts on Figures 13 and 14 into English.

The authors do not explain why the three types of time proportion (popup proportion, idle proportion, browsing proportion) are set as (0.8, 0.1, 0.1), (0.1, 0.8, 0.1), and (0.1, 0.1, 0.8). What are the impacts of the time proportion to the performance?

Answer:

The three types of time proportion (popup, idle, browsing) are set as (0.8, 0.1, 0.1), (0.1, 0.8, 0.1), and (0.1, 0.1, 0.8) so as to demonstrate the impact of the recommendations by three different triggers, i.e. popup, idle, and browsing triggers. The popup triggered recommendation appears after identifying the gender and age of the users. The browsing triggered recommendation appears during the browsing of the webpages. The idle triggered recommendation appears when the user stay (or idle) on a webpage for a certain period of time. The browsing triggered recommendation recommends products from the contents the users repeatedly search and browse and thus it is the most accurate. The click rate of the products recommended by the idle triggered recommendation is the lowest because the recommendation is made through too little information of the users. The accuracy of the popup triggered recommendation is worse than that of the browsing triggered recommendation because the identification accuracy of age and gender is not accurate enough or because of the regional cultural differences.

The authors do not describe the future works of this paper.

Answer:

The future works of this paper have been added to the end of Section 7 (Conclusions).

Reviewer 2 Report

I found this paper a joy to read. The authors marked and explained every section which makes understanding a bit easier. The following are my takes to make improvements to the paper:

1- The first paragraph of the Introduction section can be improved. There are several buzz words in it that are effective in a commercial, not in a research paper.

2- The introduction section talks about edge computing calling it a brand new innovation. To the best of this reviewer's knowledge, edge computing has been around for some time now, and actually, the paper can use some more citations to the area as well.

3- When we train, retrain, or finetune a deep learning model lots of hyperparameter setup needs to happen. I did not see the authors pay much attention to this fact. Moreover, it is unclear to me that how the data for training is provided. In the study discussed in the paper, I can see that it should be easy to gather the ground truth, but is that scaleable to every scenario? what happens if human input is needed?

4- Open Vino as we know has limitations. Such as the custom layer translation. In a general platform (which seems that the authors are presenting), how do we overcome these issues? or are we just relying on the open vino to evolve?

Author Response

We have revised the paper carefully according the reviewers’ comments. Each of the comment has been responded carefully. We have proofread the paper and revised many sentences and typos in this paper. Hence, we cannot highlight every minor revision sentence and words. Otherwise, more than a third of the sentences in this paper will be highlighted. Therefore, only the major revision parts have been highlighted.

Reviewer 2:

I found this paper a joy to read. The authors marked and explained every section which makes understanding a bit easier.

Answer:

Thank you for your appreciation.

The following are my takes to make improvements to the paper:

The first paragraph of the Introduction section can be improved. There are several buzz words in it that are effective in a commercial, not in a research paper.

Answer:

We have rewritten the first paragraph of the Introduction so as to avoid any buzz words.

The introduction section talks about edge computing calling it a brand new innovation. To the best of this reviewer's knowledge, edge computing has been around for some time now, and actually, the paper can use some more citations to the area as well.

Answer:

We agree with you. We have removed this sentence and have cited and discussed four more papers in this paper.

When we train, retrain, or finetune a deep learning model lots of hyperparameter setup needs to happen. I did not see the authors pay much attention to this fact. Moreover, it is unclear to me that how the data for training is provided. In the study discussed in the paper, I can see that it should be easy to gather the ground truth, but is that scaleable to every scenario? what happens if human input is needed?

Answer:

  • The goal of this paper is to build an AI-Edge platform which provides developer a whole deep learning development environment to setup their deep learning life cycle process. Hence, the setup of the hyperparameters is the job of the developer. That is why we did not pay much attention to the setup of the hyperparameters. The proposed AI-QSR KIOSK system is just used to demonstrate that we can build an effective AI-based recommendation system through the proposed AI-Edge platform (MiniDeep).
  • In our system, the machine itself is a training data collector. Data synchronizer is a local data storage management system which has the ability to handle the data uploading process from the edge to the cloud, and as an online training data uploading endpoint.
  • We agree with you that there will be a scalability problem if human input is needed for gathering the training data.

OpenVino as we know has limitations. Such as the custom layer translation. In a general platform (which seems that the authors are presenting), how do we overcome these issues? or are we just relying on the open vino to evolve?

Answer:

We know that OpenVino has its limitations. However, we aim to use a non-Nvidia (mainstream) architecture. We want to confirm that using OpenVino plus Intel Movidius is more suitable for an environment with small space, no fan, and less maintenance.

Reviewer 3 Report

•The language usage throughout this paper need to be improved, the author should do some proofreading on it. Give the article a mild language revision to get rid of few complex sentences that hinder readability and eradicate typo error

•Overall, the basic background is not introduced well, where the notations are not illustrated much clear. I recommend the authors to employ certain intuitive examples to elaborate the essential notations.

•Spell out each acronym the first time used in the body of the paper. Spell out acronyms in the Abstract.

•The abstract can be rewritten to be more meaningful. The authors should add more details about their final results in the abstract. Abstract should clarify what is exactly proposed (the technical contribution) and how the proposed approach is validated.

•What is the motivation of the proposed work? Research gaps, objectives of the proposed work should be clearly justified. The authors should consider more recent research done in the field of their study. Such as Industrial Cyber-Physical Systems-based Cloud IoT Edge for Federated Heterogeneous Distillation; Heuristic edge server placement in industrial internet of things and cellular networks.

·      Conclusion should state scope for future work.

·      Results need more explanations. Additional analysis is required at each experiment to show the its main purpose.

Author Response

We have revised the paper carefully according the reviewers’ comments. Each of the comment has been responded carefully. We have proofread the paper and revised many sentences and typos in this paper. Hence, we cannot highlight every minor revision sentence and words. Otherwise, more than a third of the sentences in this paper will be highlighted. Therefore, only the major revision parts have been highlighted.

Reviewer 3:

The language usage throughout this paper need to be improved, the author should do some proofreading on it. Give the article a mild language revision to get rid of few complex sentences that hinder readability and eradicate typo error.

Answer:

Thanks for your insightful comments. We have proofread the paper carefully so as to eradicate typo error. We have also rewrite many sentences of this paper so as to improve the readability of this paper.

Overall, the basic background is not introduced well, where the notations are not illustrated much clear. I recommend the authors to employ certain intuitive examples to elaborate the essential notations.

Answer:

Thanks for your insightful comment. We have rewrite many sentences of this paper so as to improve the readability of this paper.

Spell out each acronym the first time used in the body of the paper. Spell out acronyms in the Abstract.

Answer:

Thanks for your insightful comment. We have spelled out each acronym in the abstract and in the body of the paper.

The abstract can be rewritten to be more meaningful. The authors should add more details about their final results in the abstract. Abstract should clarify what is exactly proposed (the technical contribution) and how the proposed approach is validated.

Answer:

Thanks for your insightful comment. We have revised the abstract so that it would be more meaningful. We have added more details about the final results in the abstract. We also have clarified the technical contribution of the proposed scheme.

What is the motivation of the proposed work? Research gaps, objectives of the proposed work should be clearly justified. The authors should consider more recent research done in the field of their study. Such as Industrial Cyber-Physical Systems-based Cloud IoT Edge for Federated Heterogeneous Distillation; Heuristic edge server placement in industrial internet of things and cellular networks.

Answer:

Thanks for your insightful suggestion. We have added a “motivations” section to described the motivations of this paper. Besides, four more recent researches done in the field are cited and discussed in this paper, including the two papers mentioned in this comment, i.e. “Industrial Cyber-Physical Systems-based Cloud IoT Edge for Federated Heterogeneous Distillation” and “Heuristic edge server placement in industrial internet of things and cellular networks”. After reading the above two papers, we find that the goals of the above two papers are quite different from our scheme.

Conclusion should state scope for future work.

Answer:

Thanks for your insightful suggestion. The future works of this paper have been added to the end of Section 7 (Conclusions).

Results need more explanations. Additional analysis is required at each experiment to show the its main purpose.

Answer:

Thanks for your insightful comment. We have added more explanations for the results. Additional analysis for the experiment has also been added.

Reviewer 4 Report

Overall, the paper is good

-- please improve your introduction section by adding motivation and background

-- also, improve the presentation of contributions in the introduction section

-- Figures 3-4, 6-6 are complex, try to make them simple, if possible

--the conclusion is too short

-- please add some findings (in %) in the conclusion section

Author Response

We have revised the paper carefully according the reviewers’ comments. Each of the comment has been responded carefully. We have proofread the paper and revised many sentences and typos in this paper. Hence, we cannot highlight every minor revision sentence and words. Otherwise, more than a third of the sentences in this paper will be highlighted. Therefore, only the major revision parts have been highlighted.

Reviewer 4:

Overall, the paper is good

Answer:

Thank you for your appreciation.

Please improve your introduction section by adding motivation and background

also, improve the presentation of contributions in the introduction section

Answer:

Thank you for your insightful comment. We have improved the writing of the introduction. The necessity and the importance of developing the AI-Edge platform have been described and discussed (background and motivations). The contributions of the proposed scheme have also been described.

Figures 3-4, 6-6 are complex, try to make them simple, if possible

Answer:

There is no figure whose id is 3-4 or 6-6. Hence, we cannot revise the mentioned figures. Moreover, we have drawn every figures in this paper carefully so that the authors can explain the proposed scheme more clearly and the reader can read this paper more easily.

The conclusion is too short, please add some findings (in %) in the conclusion section.

Answer:

Thank you for your insightful suggestions. We have expanded the conclusion section by adding our findings and future works in this section.

Round 2

Reviewer 3 Report

none